# The proteasome biogenesis regulator Rpn4 cooperates with the unfolded protein response to promote ER stress resistance

**Rolf M Schmidt[1], Julia P Schessner[2], Georg HH Borner[2], Sebastian Schuck[1]***

[1]Center for Molecular Biology of Heidelberg University (ZMBH), DKFZ-ZMBH Alliance and CellNetworks Cluster of Excellence, Heidelberg, Germany; [2]Department of Proteomics and Signal Transduction, Max Planck Institute of Biochemistry, Martinsried, Germany

**Abstract** Misfolded proteins in the endoplasmic reticulum (ER) activate the unfolded protein response (UPR), which enhances protein folding to restore homeostasis. Additional pathways respond to ER stress, but how they help counteract protein misfolding is incompletely understood. Here, we develop a titratable system for the induction of ER stress in yeast to enable a genetic screen for factors that augment stress resistance independently of the UPR. We identify the proteasome biogenesis regulator Rpn4 and show that it cooperates with the UPR. Rpn4 abundance increases during ER stress, first by a post-transcriptional, then by a transcriptional mechanism. Induction of *RPN4* transcription is triggered by cytosolic mislocalization of secretory proteins, is mediated by multiple signaling pathways and accelerates clearance of misfolded proteins from the cytosol. Thus, Rpn4 and the UPR are complementary elements of a modular cross-compartment response to ER stress.
DOI: https://doi.org/10.7554/eLife.43244.001

*For correspondence:
s.schuck@zmbh.uni-heidelberg.de

**Competing interests:** The authors declare that no competing interests exist.

## Introduction

Cells continuously produce a large variety of proteins. To fulfill their functions, these proteins need to be properly folded, post-translationally modified, assembled into complexes and delivered to their final subcellular destinations. If errors occur in these maturation steps, compartment-specific quality control machineries clear the resulting misfolded or mislocalized proteins through refolding or degradation. These machineries are regulated by specialized adaptive responses, which adjust the folding and degradation capacity of particular organelles to ensure efficient elimination of aberrant proteins. If unresolved, protein misfolding in one compartment can disrupt overall cell homeostasis and threaten survival.

Protein maturation in the endoplasmic reticulum (ER), particularly in budding yeast, has served as a powerful paradigm to elaborate these general principles (*Patil and Walter, 2001*; *Barlowe and Miller, 2013*; *Berner et al., 2018*). Newly synthesized polypeptides containing ER sorting information, for example N-terminal signal sequences, are recognized by targeting factors such as signal recognition particle (SRP) and inserted into the ER (*Aviram and Schuldiner, 2017*). The process of protein insertion into the ER, termed translocation, is mediated by channel-forming translocon complexes (*Rapoport, 2007*). Translocation additionally requires ER-lumenal chaperones such as the highly abundant Kar2, which therefore have dual roles in protein import and folding (*Gething, 1999*; *Young et al., 2001*). Other components of the ER-resident folding machinery include oxidoreductases and glycosyltransferases (*Braakman and Hebert, 2013*). Proteins that have attained their

native conformation are sorted into ER-to-Golgi transport carriers. Proteins that fail to fold properly become subject to ER-associated degradation (ERAD). Key steps in ERAD are protein retrotranslocation into the cytosol with the help of the AAA ATPase Cdc48, followed by degradation by the proteasome (*Berner et al., 2018*). If these mechanisms are insufficient, misfolded proteins accumulate. This condition constitutes ER stress and activates an adaptive response called the unfolded protein response (UPR) (*Walter and Ron, 2011*). The UPR is triggered when misfolded proteins are sensed by the ER transmembrane protein Ire1. Activated Ire1 initiates non-conventional cytosolic splicing of the *HAC1* mRNA, enabling production of the transcription factor Hac1. In turn, Hac1 induces numerous genes involved in ER function (*Travers et al., 2000*). The resulting increase in ER protein folding capacity resolves ER stress, closing a homeostatic feedback loop. The physiological significance of the UPR is demonstrated by yeast mutants lacking Ire1 or Hac1. When challenged by ER stress, these mutants exhibit a variety of defects in translocation, glycosylation, ERAD and ER-to-Golgi transport, and rapidly lose viability (*Cox et al., 1993*; *Spear and Ng, 2003*).

A number of UPR-independent pathways respond to, and help mitigate, ER stress. These pathways include MAP kinase signaling through Slt2/Mpk1 and Hog1, the Hsf1-dependent heat shock response and protein kinase A (PKA) signaling (*Bonilla and Cunningham, 2003*; *Chen et al., 2005*; *Liu and Chang, 2008*; *Bicknell et al., 2010*; *Hou et al., 2014*; *Pincus et al., 2014*). However, exactly how they counteract ER stress has been difficult to determine. For instance, ER stress is alleviated by augmented ER-to-Golgi transport and enhanced elimination of reactive oxygen species downstream of the heat shock response and also by reduced protein synthesis downstream of PKA signaling (*Liu and Chang, 2008*; *Hou et al., 2014*; *Pincus et al., 2014*). Yet, these mechanisms only partially explain the beneficial effects of the signaling pathways controlling them. Finally, the UPR can, by unknown means, be amplified by Ire1-independent induction of *HAC1* transcription (*Leber et al., 2004*). Therefore, it remains to be fully defined which pathways cooperate with the UPR and how they contribute to ER stress resistance.

Here, we identify the proteasome biogenesis regulator Rpn4 as an important UPR-independent factor that promotes resistance to ER stress in yeast. We show that protein misfolding induces Rpn4 activity by post-transcriptional and transcriptional mechanisms, and provide evidence that Rpn4 complements the UPR by enhancing protein quality control in the cytosol.

## Results

### A titratable system for the induction of ER stress

To identify pathways cooperating with the UPR, we searched for genes that can augment resistance to ER stress in UPR-deficient cells. Mutants lacking Ire1 or Hac1 grow normally under optimal conditions but cannot proliferate under even mild ER stress (*Cox et al., 1993*; *Spear and Ng, 2003*; *Schuck et al., 2009*). We hypothesized that UPR mutants can be protected against ER stress by overexpression of genes that complement the UPR. If so, such genes should be identifiable through a screen based on cell growth phenotypes.

To implement this idea, we established a titratable system for the induction of ER stress. We used CPY*, a folding-defective variant of the soluble vacuolar carboxypeptidase Y (*Finger et al., 1993*). We chose an HA-tagged mutant variant of CPY* that lacks all of its four N-glycosylation sites and is here referred to as non-glycosylatable (ng) CPY*. After translocation into the ER, this variant is unable to fold properly and is neither efficiently cleared by ERAD nor exported to the Golgi complex (*Knop et al., 1996*; *Spear and Ng, 2005*; *Kostova and Wolf, 2005*). As a result, ngCPY* accumulates in the ER lumen and burdens the ER protein folding machinery. We placed ngCPY* under the control of the *GAL* promoter and the artificial transcription factor Gal4-ER-Msn2 (GEM). This system allows regulation of gene expression with the exogenous steroid β-estradiol (*Pincus et al., 2014*). Estradiol-driven expression of ngCPY* caused dose-dependent activation of the UPR as measured with a *HAC1* splicing reporter (*Figure 1A*; *Pincus et al., 2010*). *HAC1* splicing obtained with ≥100 nM estradiol was similar to that elicited by 0.5 μg/ml of the general ER stressor tunicamycin (*Figure 1—figure supplement 1A*). For all estradiol concentrations tested, *HAC1* splicing declined at later time points, suggesting that cells adapted to the stress and inactivated the UPR. Estradiol-induced expression of glycosylatable CPY*, which is degraded through ERAD and can be exported from the ER, yielded much weaker and more transient UPR activation (*Figure 1—figure supplement*

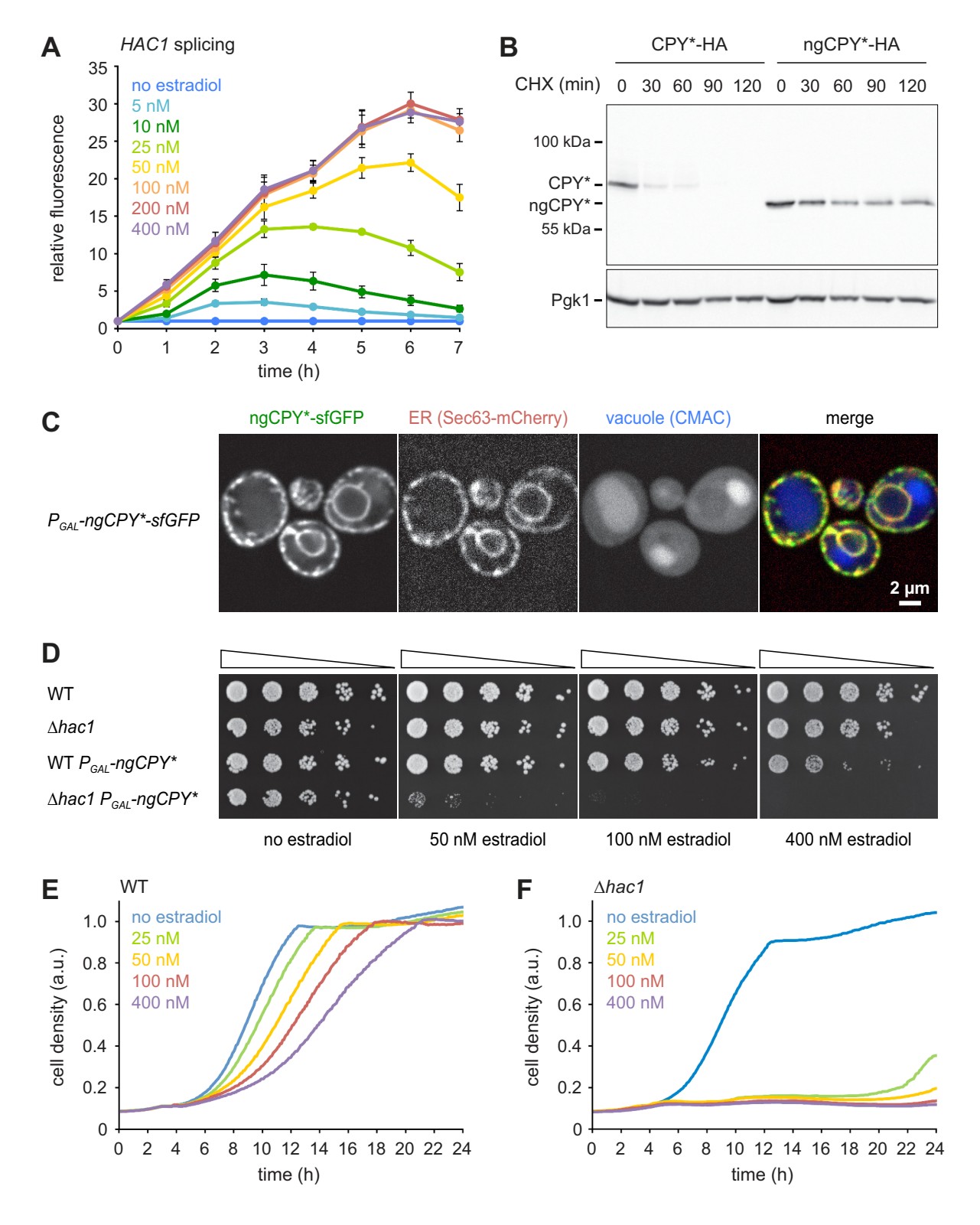

**Figure 1.** A titratable system for the induction of ER stress. (A) Flow cytometric measurement of GFP levels in cells harboring the *HAC1* splicing reporter and expressing ngCPY* under the control of the estradiol-inducible *GAL* promoter system. For each time point, data are normalized to untreated cells. Mean ±SEM, n = 3. (B) Western blot of HA and Pgk1 from cells expressing CPY*-HA or ngCPY*-HA. Cells were treated with cycloheximide (CHX) for the times indicated. Pgk1 served as a loading control. (C) Images of cells expressing ngCPY*-sfGFP and the general ER marker

*Figure 1 continued on next page*

*Figure 1 continued*

Sec63-mCherry. Expression of ngCPY*-sfGFP was induced with 25 nM estradiol for 4 hr and cells were stained with the vacuole dye CMAC. (D) Growth assay on solid media of wild-type (WT) and Δhac1 cells expressing the estradiol-inducible artificial transcription factor GEM and, where indicated, ngCPY* under the control of the *GAL* promoter. For each strain, series represent fivefold dilution steps. (E) Growth assay in liquid media of WT cells expressing ngCPY* under the control of the estradiol-inducible *GAL* promoter system. a.u., arbitrary units. (F) As in panel E, but with Δhac1 cells.
DOI: https://doi.org/10.7554/eLife.43244.002
The following figure supplement is available for figure 1:

**Figure supplement 1.** A titratable system for the induction of ER stress.
DOI: https://doi.org/10.7554/eLife.43244.003

*1B*). Cycloheximide chase experiments confirmed that ngCPY* was substantially more resistant to degradation than glycosylatable CPY*, as reported (*Figure 1B*; *Knop et al., 1996*). Furthermore, ngCPY* tagged with superfolder GFP (sfGFP) was largely retained in the ER (*Figure 1C*). In contrast, CPY*(N479Q)-sfGFP, which lacks the glycosylation site required for efficient ERAD but contains the three glycosylation sites needed for ER export, escaped to the vacuole (*Figure 1—figure supplement 1C*; *Kawaguchi et al., 2010*).

Estradiol did not affect growth of wild-type or Δhac1 cells expressing only the artificial transcription factor GEM (*Figure 1D*). Furthermore, estradiol-induced expression of ngCPY* in wild-type cells caused only modest growth defects on solid media, even at the highest estradiol concentration tested. In Δhac1 cells, however, induction of ngCPY* expression with increasing concentrations of estradiol strongly retarded and eventually prevented cell proliferation. Growth assays in liquid media yielded similar results (*Figure 1E and F*). To determine whether the lack of proliferation of Δhac1 cells reflected a growth arrest or cell death, we assayed cell viability after expression of ngCPY*. No loss of viability of Δhac1 cells occurred, even with estradiol concentrations that completely blocked proliferation (*Figure 1—figure supplement 1D*). Hence, expression of ngCPY* merely caused a growth arrest. In summary, this estradiol-controlled system can be titrated to induce defined levels of ER stress and can be used to prevent growth of UPR-deficient cells.

## A screen for genes promoting ER stress resistance in UPR mutants

We exploited the ngCPY*-induced growth arrest in UPR mutants to conduct a multicopy suppression screen. Cells lacking *HAC1* and containing the titratable ER stress system were transformed with a genomic library in a high-copy vector. Transformants were first allowed to form colonies on estradiol-free solid medium and then replicated onto media containing a range of estradiol concentrations sufficient to block growth of the parental strain. Plasmids were retrieved from transformants able to grow on estradiol-containing plates and their inserts were sequenced. The obtained candidate genes were individually subcloned into the same high-copy vector and tested for their ability to suppress the toxicity of ngCPY* in Δhac1 cells.

As expected, the strongest suppressor was *HAC1*, which complemented the *HAC1* deletion in the parental strain. The next strongest suppressors as judged by growth assays on solid and in liquid media were *RPN4*, *PDR1* and *SSZ1*. Additional, but weaker, suppressors were *YAP1*, *CAD1*, *SAF1*, *MUM2* and *NOP56* (*Table 1* and *Figure 2*). Rpn4 is a transcription factor that localizes to the cytosol and the nucleus (*Tkach et al., 2012*). Its primary function is the induction of genes encoding proteasome subunits (*Dohmen et al., 2007*). Pdr1 is another transcription factor, controls genes involved in multidrug resistance and induces *RPN4* transcription (*Owsianik et al., 2002*; *Prasad and Goffeau, 2012*). Ssz1 is part of the ribosome-associated chaperone complex and its overproduction activates Pdr1 (*Hallstrom et al., 1998*; *Conz et al., 2007*). Yap1 is a third transcription factor, mediates the oxidative stress response and also induces *RPN4* transcription (*Owsianik et al., 2002*). Considering these links, we tested whether Pdr1, Ssz1 and Yap1 enhanced ER stress resistance by raising Rpn4 levels. We generated Δhac1 cells in which the *RPN4* gene was controlled by the constitutive *CYC1* promoter and thus uncoupled from its normal regulation. We then overexpressed *RPN4*, *PDR1*, *SSZ1* or *YAP1* and assayed growth in the presence ngCPY*. Overexpression of all four genes promoted growth under these conditions (*Figure 3A*). Hence, Pdr1, Ssz1 and Yap1 can enhance ER stress resistance also when they are unable to activate the *RPN4* promoter. It therefore appears likely that Pdr1 and Yap1 relieve ER stress in our system by inducing target genes other than *RPN4*.

**Table 1.** Genes identified in the screen.

| Gene | Description |
| --- | --- |
| HAC1 | transcription factor, regulates the unfolded protein response |
| RPN4 | transcription factor, stimulates expression of proteasome genes |
| PDR1 | transcription factor, regulates the pleiotropic drug response, induces RPN4 transcription |
| SSZ1 | Hsp70 chaperone, part of ribosome-associated complex, overexpression activates Pdr1 |
| MUM2 | protein essential for meiotic DNA replication and sporulation |
| YAP1 | transcription factor, regulates response to oxidative stress, induces RPN4 transcription |
| CAD1 | transcription factor, involved in stress responses, paralog of YAP1 |
| SAF1 | F-box protein, subunit of SCF ubiquitin ligase complexes |
| NOP56 | essential nucleolar protein |

DOI: https://doi.org/10.7554/eLife.43244.006

In the case of Pdr1, candidates include genes for multidrug transporters such as *PDR5*, *PDR10* and *PDR15*, which may export estradiol from cells and thereby lower the levels of ngCPY*.

The identification of Rpn4 suggests that enhanced proteasome biogenesis promotes ER stress resistance. However, the entirety of genes controlled by Rpn4, here referred to as the Rpn4 regulon, encompasses many genes beyond those encoding proteasome subunits (*Mannhaupt et al., 1999*; *Jelinsky et al., 2000*). Rpn4 may therefore counteract ER stress by more than one mechanism. Indeed, the *RPN4* regulator Yap1 is itself activated by Rpn4 (*Mannhaupt et al., 1999*) and may aid stress resistance by preventing oxidative damage. Furthermore, we directly tested the Rpn4 target gene *CDC48* (*Bosis et al., 2010*), even though it was not found in the screen. Overexpression of *CDC48* also restored growth of Δ*hac1* cells expressing ngCPY*, although weakly compared to over-expression of *RPN4* (*Figure 2B-E*). Overall, the outcome of the screen suggests that Rpn4 activity becomes limiting for cell proliferation during ER stress, at least in UPR mutants.

## Rpn4 and the UPR cooperate to counteract ER stress

*RPN4* is not controlled by Hac1 and hence not a UPR target gene (*Travers et al., 2000*; *Pincus et al., 2014*). This lack of a direct link suggests that Rpn4 acts in parallel to, rather than downstream of, the UPR (*Ng et al., 2000*). To better understand the interplay between Rpn4 and the UPR, we manipulated Rpn4 levels in wild-type and Δ*hac1* cells and challenged these cells with ngCPY* or tunicamycin (*Figure 3B and C*). Overexpression of *RPN4* in wild-type cells increased resistance to both stressors. This observation indicates that Rpn4 activity becomes limiting for cell proliferation during ER stress also when the UPR is intact. Conversely, deletion of *RPN4* caused a general growth defect and sensitized cells to ngCPY* and tunicamycin, consistent with previous reports (*Wang et al., 2008*; *Hou et al., 2014*; *Rousseau and Bertolotti, 2016*). As expected, Δ*hac1* cells were highly vulnerable to either ER stressor, and *RPN4* overexpression provided a measure of protection. Δ*hac1* Δ*rpn4* cells grew very poorly already in the absence of ngCPY* or tunicamycin, and not at all in their presence. The synthetic sickness phenotype of Δ*hac1* Δ*rpn4* cells agrees with earlier reports and shows that Rpn4 and the UPR are functionally linked (*Ng et al., 2000*; *Hou et al., 2014*).

The slow growth of Δ*hac1* Δ*rpn4* double mutants compared with Δ*rpn4* single mutants implies that the UPR is constitutively active and physiologically important in Rpn4-deficient cells. Indeed, *HAC1* splicing was elevated in untreated Δ*rpn4* cells, consistent with high throughput data (*Figure 4A*; *Jonikas et al., 2009*). Accordingly, quantitative real-time PCR showed that mRNA levels of the UPR target genes *KAR2* and *SIL1* were higher in Δ*rpn4* mutants than in wild-type cells (*Figure 4B and C*). To further characterize the effect of *RPN4* deletion, we compared the proteomes of wild-type and Δ*rpn4* cells by quantitative mass spectrometry. As expected, the abundance of proteasome subunits was reduced in cells lacking Rpn4, while the abundance of proteins encoded by UPR target genes was increased (*Figure 4D* and *Figure 4—source data 1*). Many other proteins were changed significantly, illustrating the profound impact of loss of Rpn4. First, Cdc48 levels were lower in Δ*rpn4* cells. Second, the levels of the stress-inducible proteasome assembly chaperone Tma17 and its activator Slt2/Mpk1 were elevated (*Hanssum et al., 2014*; *Rousseau and Bertolotti,*

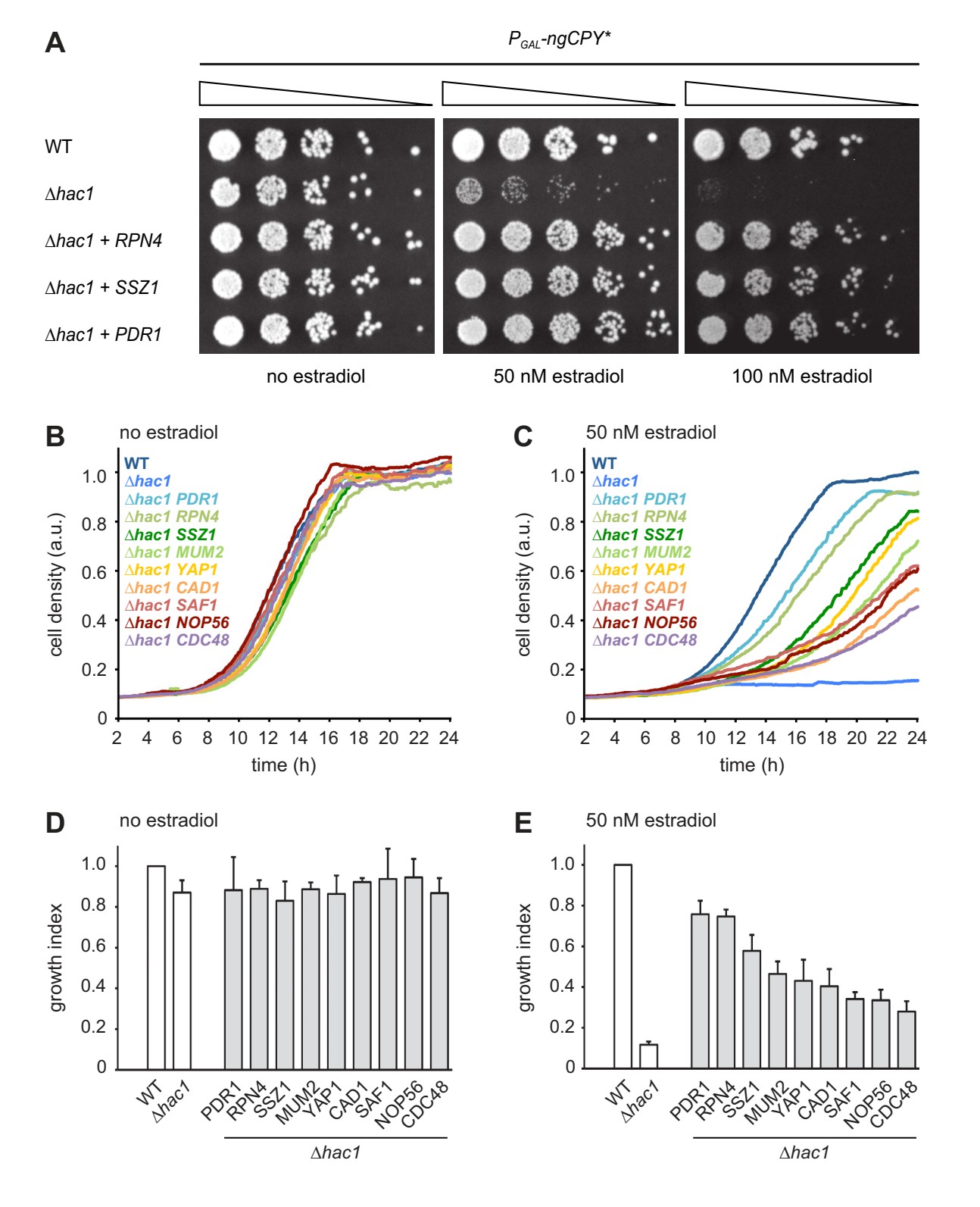

**Figure 2.** Multicopy suppression of ngCPY* toxicity in Δhac1 cells. (A) Growth assay on solid media of wild-type (WT) and Δhac1 cells expressing ngCPY* under the control of the estradiol-inducible *GAL* promoter system and overexpressing *RPN4*, *SSZ1* or *PDR1* from extrachromosomal plasmids where indicated. (B) Growth assay in liquid medium of WT and Δhac1 cells expressing ngCPY* under the control of the estradiol-inducible *GAL* promoter system. Δhac1 cells additionally overexpressed the indicated genes. Cells were grown without estradiol. a.u., arbitrary units. (C) As in panel B,
*Figure 2 continued on next page*

*Figure 2 continued*

but in the presence of 50 nM estradiol. (D) Quantification of growth assays as shown in panel B. Data are normalized to WT cells. Mean ±SEM, n = 4. (E) Quantification of growth assays as shown in panel C. Data are normalized to WT cells. Mean ±SEM, n = 4.

DOI: https://doi.org/10.7554/eLife.43244.004

*2016*). Third, cytosolic protein folding machinery was upregulated and the abundance of many ribosomal proteins was suppressed, indicating a broad stress response (*Gasch et al., 2000*). Fourth, many proteins that are involved in sterol synthesis and constitutively turned over by different ERAD pathways were more abundant in Δ*rpn4* cells. These proteins included Erg1, 3, 5, 9, 11, 25, 27, 28 and Hmg2, although only Erg1 and Erg3 met our stringent criteria for significance (*Hampton et al., 1996*; *Foresti et al., 2013*; *Foresti et al., 2014*; *Hitchcock et al., 2003*; *Khmelinskii et al., 2014*; *Christiano et al., 2014*). Their accumulation implies generally impaired ERAD in Δ*rpn4* cells. Importantly, neither Erg1 nor Erg3 are regulated by the UPR (*Travers et al., 2000*; *Pincus et al., 2014*), arguing against the possibility that their increased abundance can be explained by the constitutive activation of the UPR in Δ*rpn4* cells. In sum, cells lacking Rpn4 suffer from chronic ER stress, likely because inefficient ERAD leads to an accumulation of misfolded and redundant proteins in the ER. In response, they activate the UPR, augment proteasome assembly capacity, enhance cytosolic protein folding and attenuate protein synthesis.

The above results show that cells attempt to compensate the lack of Rpn4 by activating the UPR. To test whether the inverse is true and cells compensate the lack of a functionally sufficient UPR by activating the Rpn4 regulon, we monitored Rpn4 abundance. Rpn4 is a short-lived protein that is rapidly turned over by the proteasome (*Xie and Varshavsky, 2001*). When proteasome capacity becomes limiting, Rpn4 accumulates and promotes proteasome biogenesis until its degradation is restored. This negative feedback loop homeostatically adjusts proteasome activity (*Dohmen et al., 2007*). The levels of chromosomally tagged Rpn4-HA in wild-type and Δ*hac1* cells were similar at steady state (*Figure 5A and B*). Upon treatment with 2 µg/ml tunicamycin, they increased two-fold within 15 min. Subsequently, Rpn4 abundance continued to climb but did so more quickly in Δ*hac1* cells, reaching more than four-fold basal levels after 60 min. To determine whether this rise involved an upregulation of *RPN4* transcription, we analyzed *RPN4* mRNA by quantitative real-time PCR. In wild-type cells, tunicamycin treatment for up to 60 min induced the UPR target genes *KAR2* and *SIL1* (*Figure 5C* and *Figure 5—figure supplement 1*), but not *RPN4* (*Figure 5D*). Therefore, under these conditions, Rpn4 accumulates through a post-transcriptional mechanism, presumably by slowed degradation. In Δ*hac1* cells, tunicamycin treatment induced *KAR2* and *SIL1* less strongly than in wild-type cells but increased *RPN4* mRNA levels. While we cannot rule out changes in mRNA stability, we assume that this rise reflects enhanced transcription. This response likely contributes to the more pronounced stress-induced upregulation of Rpn4 abundance in Δ*hac1* cells. Importantly, prolonged ER stress upon treatment with 5 µg/ml tunicamycin raised *RPN4* mRNA levels also in wild-type cells (*Figure 5E*). Accordingly, the protein abundance of proteasome subunits and all detectable proteasome assembly chaperones was elevated (*Figure 5F* and *Figure 4—source data 1*). Hence, during ER stress, cells augment the UPR by enhancing Rpn4 activity and promoting proteasome biogenesis. Interestingly, proteasome subunit abundance still increased during ER stress in Δ*rpn4* cells (*Figure 5—figure supplement 2*), possibly reflecting a recently suggested post-transcriptional mechanism mediated by Slt2/Mpk1 (*Rousseau and Bertolotti, 2016*). Nevertheless, the levels of proteasome subunits in ER-stressed Δ*rpn4* cells remained below even those of untreated wild-type cells. Collectively, these results show that activation of the UPR and the Rpn4 regulon are two cooperating elements of the cellular response to ER stress.

## Rpn4 is upregulated by and protects against mislocalized secretory proteins

We next asked how ER stress upregulates Rpn4. The stronger increase in Rpn4 abundance and the more sensitive *RPN4* transcriptional response in Δ*hac1* cells may reflect more severe ER stress in these mutants. To test this assumption, we used protein translocation as readout for ER function. When the capacity of ER chaperones is exhausted, they can no longer assist protein import. As a result, translocation is compromised, including that of Kar2 itself (*Vogel et al., 1990*). Western

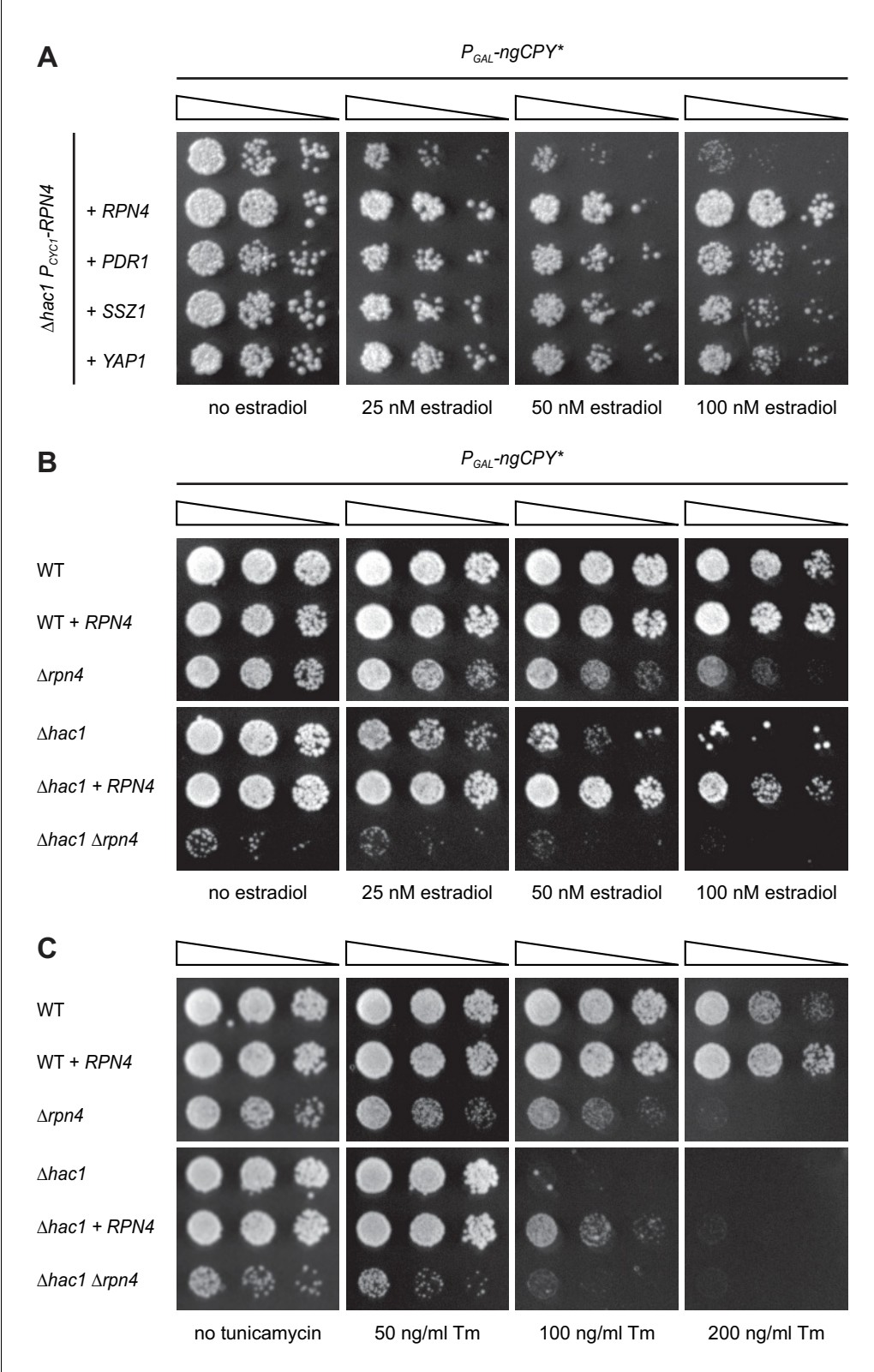

**Figure 3.** Rpn4 and the UPR are functionally linked. (**A**) Growth assay on solid media of Δ*hac1* cells containing the constitutive *CYC1* promoter in place of the endogenous *RPN4* promoter, expressing ngCPY* under the control of the estradiol-inducible *GAL* promoter system and overexpressing *RPN4*, *SSZ1*, *PDR1* or *YAP1* where indicated. (**B**) Growth assay on solid media of wild-type (WT), Δ*rpn4*, Δ*hac1* and Δ*hac1* Δ*rpn4* cells expressing ngCPY* under

*Figure 3 continued on next page*

*Figure 3 continued*

the control of the estradiol-inducible *GAL* promoter system and overexpressing *RPN4* where indicated. (**C**) As in panel B, but on media containing different concentrations of tunicamycin (Tm).

DOI: https://doi.org/10.7554/eLife.43244.005

blotting showed a single band for Kar2 in untreated wild-type and Δ*hac1* cells (*Figure 6A*). Treatment with 2 µg/ml tunicamycin for up to 60 min did not change Kar2 levels. However, a second Kar2 band of slightly higher molecular weight appeared in Δ*hac1* cells, starting at 30 min. Since Kar2 is not glycosylated, this upshift indicates retention of its cleavable signal sequence (ss) and suggests that the slower migrating form corresponds to untranslocated ss-Kar2 (*Ng et al., 1996*). Hence, under these conditions, Δ*hac1* cells show signs of overwhelmed chaperone capacity. Treatment with 5 µg/ml tunicamycin increased Kar2 abundance in wild-type cells and caused the appearance of ss-Kar2 in both strains (*Figure 6B*). Therefore, strong ER stress impairs translocation also in wild-type cells. These results confirm that ER stress is initially buffered by the UPR in wild-type cells but rapidly disrupts ER function in Δ*hac1* cells.

The above results reveal a conspicuous correlation between impaired translocation and elevated *RPN4* mRNA levels. Both phenomena occur in Δ*hac1* cells exposed to 2 µg/ml tunicamycin, whereas 5 µg/ml are required in wild-type cells. Impaired translocation and increased *RPN4* mRNA levels could be unrelated consequences of ER stress. Alternatively, their correlation could reflect a causal relationship, with translocation defects activating *RPN4* transcription. To distinguish between these possibilities, we employed *sec65-1* cells, which express a temperature-sensitive variant of the SRP subunit Sec65 (*Stirling et al., 1992*). Disruption of SRP function rapidly inhibits translocation and causes accumulation of secretory proteins in the cytosol, where they are unable to fold properly. As expected, Kar2 translocation was intact in *sec65-1* cells at the permissive temperature of 25°C but impaired upon a shift to 30°C or above (*Figure 6C*; *Ng et al., 1996*). *RPN4* mRNA levels did not change when wild-type or *sec65-1* cells were shifted from 25°C to 28°C (*Figure 6—figure supplement 1*). However, shifts to temperatures of 30°C or above raised *RPN4* mRNA levels specifically in *sec65-1* cells (*Figure 6D* and *Figure 6—figure supplement 1*). Importantly, *HAC1* splicing was not activated under these conditions (*Figure 6E*). Therefore, disrupted translocation induces the *RPN4* gene even in the absence of ER stress. This finding indicates that stress-induced translocation defects activate *RPN4* transcription.

To examine the physiological significance of *RPN4* expression in cells suffering from translocation defects, we analyzed growth of *sec65-1* cells at different temperatures. Cells grew normally at up to 28°C but showed almost no growth at 30°C or above (*Figure 7A*). These observations are consistent with the described tight temperature sensitivity of the *sec65-1* allele (*Stirling et al., 1992*). *RPN4* overexpression restored some growth at 30°C and 32°C, showing that the levels of Rpn4 were physiologically important under these conditions. Next, we tested whether elevated Rpn4 abundance promoted the degradation of misfolded cytosolic proteins. Cycloheximide chase experiments showed that *RPN4* overexpression accelerated the degradation of Δss-ngCPY*-HA, which mislocalizes to the cytosol due to deletion of its signal sequence (*Figure 7B and C*). The same was true for Luciferase(DM)-mCherry, another misfolded cytosolic model protein (*Figure 7D*). Collectively, these experiments indicate that the Rpn4 regulon is activated by and protects against cytosolic mislocalization of secretory proteins.

Our results show that *RPN4* overexpression protects against ER stressors and cytosolically mislocalized secretory proteins, suggesting that it counteracts folding stress in the ER and the cytosol. To test this notion, we used reporters for three different stress response pathways: the *HAC1* splicing reporter for the UPR, a heat shock element (HSE) reporter for the Hsf1-dependent heat shock response (*Zheng et al., 2016*) and an *HSP12* reporter for the Msn2/4-dependent general stress response (*Pincus et al., 2014*). The transcription factors Msn2 and Msn4 are kept inactive by PKA (*Görner et al., 1998*). Stress conditions inhibit PKA, resulting in derepression of Msn2/4 and induction of genes such as *HSP12* (*Pincus et al., 2014*). Although Hsf1 and the PKA-Msn2/4 pathway respond to prolonged ER stress (*Liu and Chang, 2008*; *Pincus et al., 2014*), they monitor the folding environment in the cytosol, and we employed the *HSE* and *HSP12* reporters to read out cytosolic protein homeostasis. We introduced the reporters into Δ*hac1* cells that did or did not overexpress

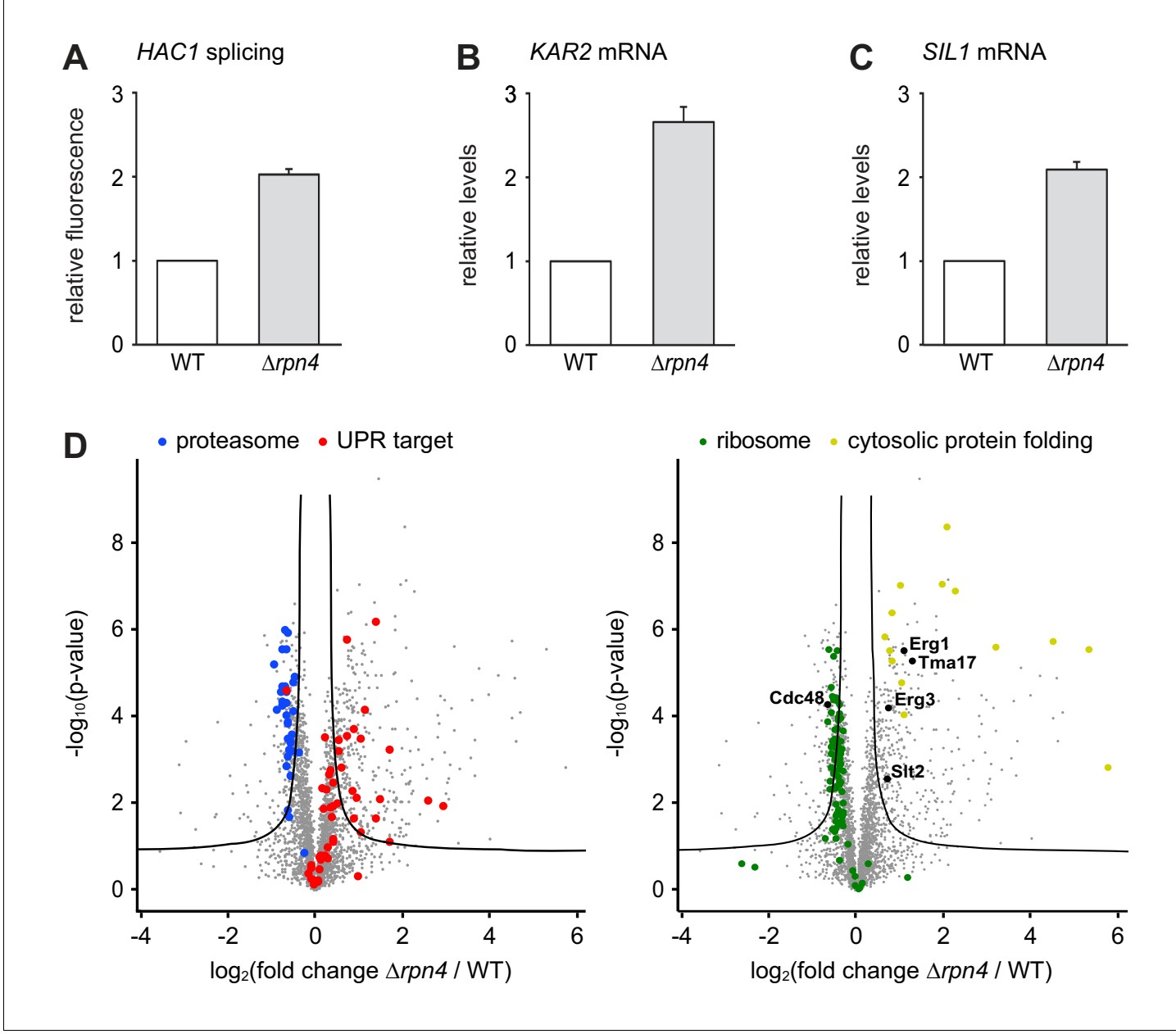

**Figure 4.** Loss of Rpn4 activates the UPR and triggers a broad adaptive response. (**A**) Flow cytometric measurement of GFP levels relative to cytosolic BFP in wild-type (WT) and Δrpn4 cells harboring the *HAC1* splicing reporter. Data are normalized to WT cells. Mean ±SEM, n = 3. (**B**) *KAR2* mRNA levels in WT and Δrpn4 cells as measured by quantitative real-time PCR. Data are normalized to WT cells. Mean ±SEM, n = 3. (**C**) As in panel B, but for *SIL1*. (**D**) Global effects of *RPN4* deletion on protein expression. For each protein, the x axis shows the average $\log_2$ fold change between WT and Δrpn4 cells (proteins increased in the Δrpn4 strain have positive values); the y axis shows the result of a t test for that difference (two-tailed; n = 4). The ''volcano'' lines indicate thresholds of significance. Proteins falling above the volcano lines are significantly changed. The left and right panels show the same plot but with different proteins highlighted. See *Figure 4—source data 1* for the data used to generate the plot. In Δrnp4 cells, proteasome subunits are downregulated (blue dots, $p=1.1\times10^{-17}$, n = 32), UPR targets are upregulated (red dots, $p=1.8\times10^{-9}$, n = 50), ribosomal proteins are downregulated (green dots, $p=7.5\times10^{-29}$, n = 89) and proteins involved in cytosolic protein folding are upregulated (yellow dots, $p=3.8\times10^{-9}$, n = 14).
DOI: https://doi.org/10.7554/eLife.43244.007

The following source data is available for figure 4:

**Source data 1.** Proteomics data.
DOI: https://doi.org/10.7554/eLife.43244.008

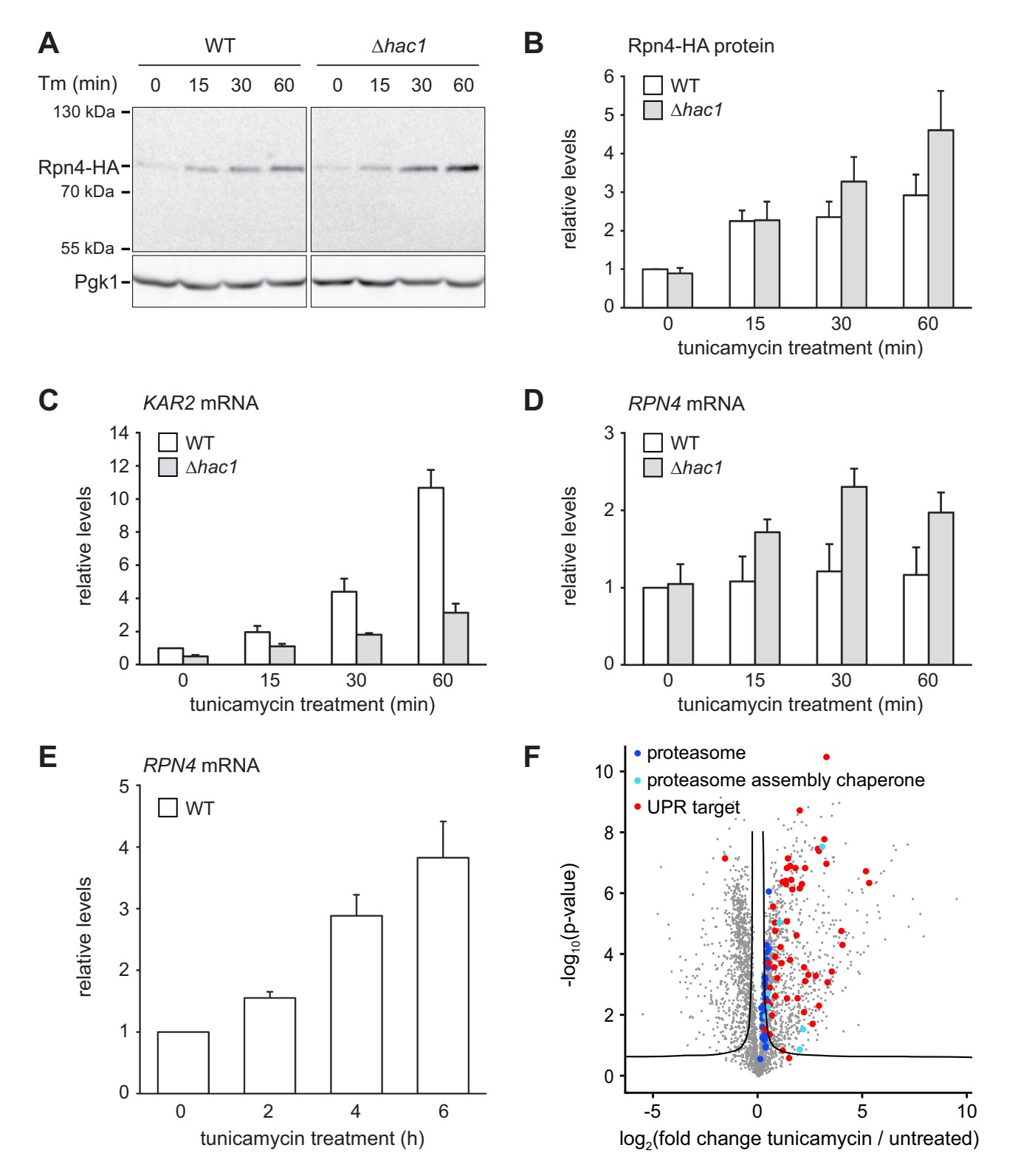

**Figure 5.** ER stress increases Rpn4 abundance, induces *RPN4* transcription and promotes proteasome biogenesis. (**A**) Western blot of HA and Pgk1 from wild-type (WT) and Δ*hac1* cells expressing Rpn4-HA and treated with 2 µg/ml tunicamycin (Tm) for the times indicated. (**B**) Quantification of Rpn4-HA protein levels relative to Pgk1 from western blots as shown in panel A. Data are normalized to WT cells at t = 0. Mean ±SEM, n = 5. (**C**) *KAR2* mRNA levels in WT and Δ*hac1* cells treated with 2 µg/ml tunicamycin for the times indicated. Data are normalized to WT cells at t = 0. Mean ±SEM, n = 3. (**D**)
*Figure 5 continued on next page*

*Figure 5 continued*

As in panel C, but for *RPN4*. (E) As in panel C, but for *RPN4* after treatment with 5 μg/ml tunicamycin for the times indicated. (F) Global effects of tunicamycin treatment on protein expression. For each protein, the x axis shows the average $\log_2$ fold change between untreated WT cells and WT cells treated with 5 μg/ml tunicamycin for 4 hr (proteins upregulated by the treatment have positive values); the y axis shows the result of a t test for that difference (two-tailed; n = 4). The ''volcano'' lines indicate thresholds of significance. Proteins falling above the volcano lines are significantly changed. See *Figure 4—source data 1* for the data used to generate the plot. Treatment with tunicamycin causes upregulation of proteasome subunits (dark blue dots, p=$7.6\times10^{-3}$, n = 32), proteasome assembly chaperones (light blue dots, p=$3.3\times10^{-3}$, n = 8), and UPR targets (red dots, p=$3.3\times10^{-22}$, n = 58).

DOI: https://doi.org/10.7554/eLife.43244.009

The following figure supplements are available for figure 5:

**Figure supplement 1.** *SIL1* mRNA levels in wild-type and Δ*hac1* cells treated with tunicamycin.

DOI: https://doi.org/10.7554/eLife.43244.010

**Figure supplement 2.** Effects of tunicamycin treatment on the levels of proteasome subunits in wild-type and Δ*rpn4* cells.

DOI: https://doi.org/10.7554/eLife.43244.011

*RPN4*, induced ER stress by ngCPY* expression and determined reporter activity. Expression of ngCPY* activated all three reporters, which was attenuated by *RPN4* overexpression (*Figure 7E*). This result shows that Rpn4 can indeed improve protein homeostasis in both the ER and the cytosol.

## Multiple signaling pathways mediate *RPN4* induction by ER stress

Finally, we investigated through which signaling pathways ER stress induces the *RPN4* gene. The *RPN4* promoter contains well-characterized binding sites for Pdr1/3, Yap1 and Hsf1, called the Pdr1/3 response element (PDRE), Yap1 response element (YRE) and heat shock element (HSE), respectively (*Owsianik et al., 2002*; *Hahn et al., 2006*). To test the relevance of these binding sites, we generated *RPN4* reporters in which different *RPN4* promoter variants controlled expression of the fast-maturing fluorescent protein mNeonGreen and measured mNeonGreen levels by flow cytometry. Steady-state activity of the *RPN4* reporter was essentially unchanged by mutation of the two PDREs or the YRE but was reduced by 40% upon mutation of the HSE (*Figure 8A*). Tunicamycin treatment activated the *RPN4* reporter, which was unaffected by mutation of any of the above promoter elements (*Figure 8B*). Hence, Hsf1 regulates basal *RPN4* activity, but Pdr1/3, Yap1 and Hsf1 appear to be individually dispensable for *RPN4* induction by ER stress.

Given that ER stress triggers the general stress response (*Pincus et al., 2014*), we asked whether Msn2/4 downstream of PKA can activate the *RPN4* gene. We used a strain in which the PKA isoforms Tpk1/2/3 had been modified such that their enlarged ATP binding pockets allowed specific inhibition of these kinases with the bulky ATP analog 1NM-PP1 (*Hao and O'Shea, 2012*). 1NM-PP1 treatment of cells harboring the analog-sensitive *tpk1/2/3-as* alleles induced both the *RPN4* reporter and the endogenous *RPN4* gene (*Figure 8C and D*). As expected, deletion of *MSN2/4* strongly reduced activation of the *RPN4* reporter upon PKA inhibition (*Figure 8C*). Activation of the *RPN4* reporter by tunicamycin was blunted in Δ*msn2/4* cells but not blocked (*Figure 8E*), suggesting that prolonged ER stress activates partially redundant mechanisms to induce *RPN4*. Accordingly, combined deletion of *MSN2/4* and mutation of the YRE or the HSE additively reduced activation of the *RPN4* reporter. Mutation of both the YRE and the HSE almost completely abolished reporter activation in a Δ*msn2/4* background. These results show that multiple signaling pathways contribute to the induction of *RPN4* transcription by ER stress.

## Discussion

We have shown that Rpn4 and the UPR cooperate to counteract ER stress. Based on our findings, we propose the following model (*Figure 8F*). Protein misfolding in the ER burdens the proteasome by increased flux through the ERAD pathway, activates the UPR and eventually impairs protein translocation into the ER. As a result, secretory proteins mislocalize to the cytosol, where they are unable to fold properly and further strain proteasome capacity. Inefficient proteasomal degradation leads to an increase in Rpn4 abundance and activation of the Rpn4 regulon, which enhances proteasome biogenesis to clear misfolded ER and cytosolic proteins. Furthermore, persistent cytosolic protein

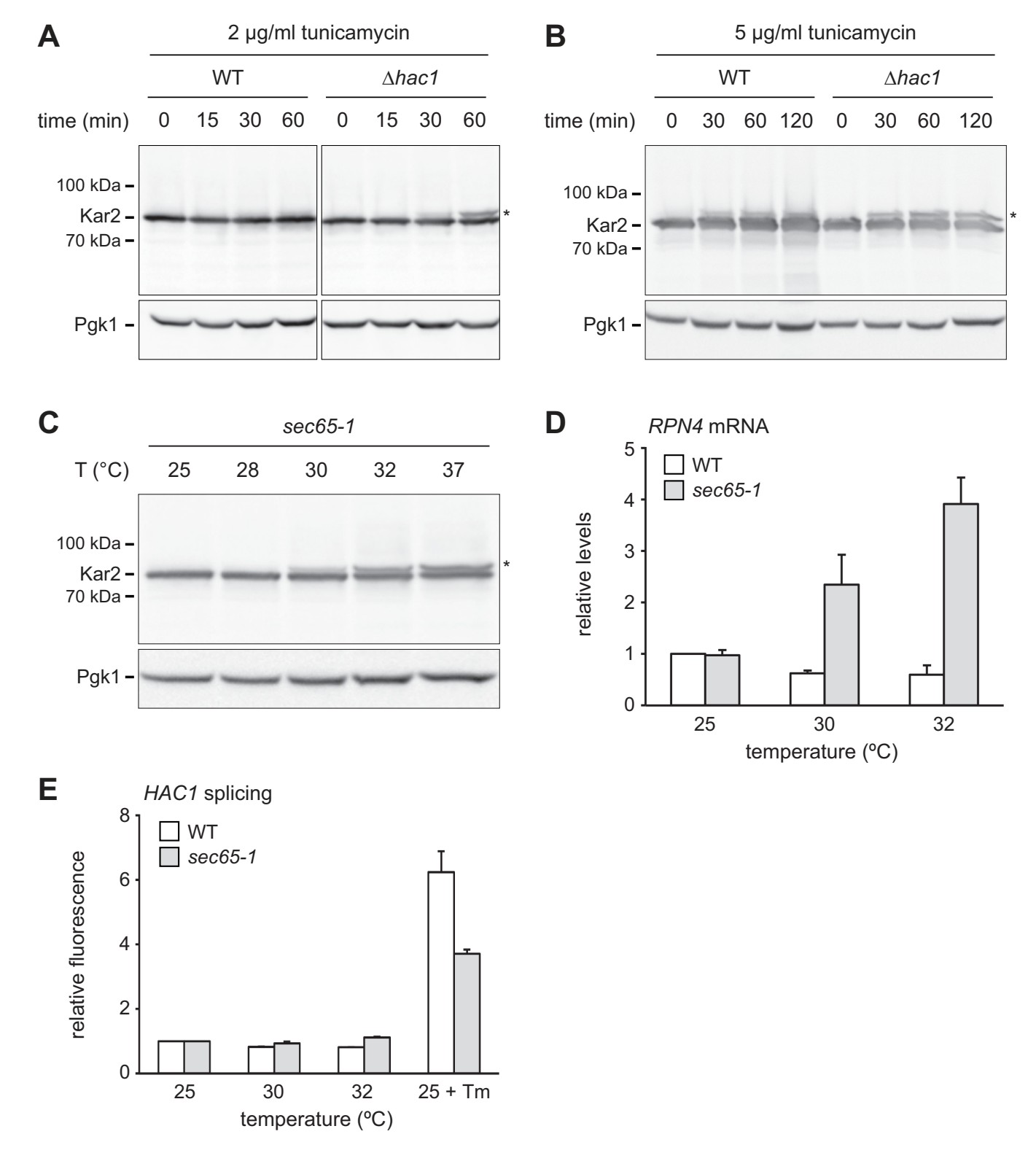

**Figure 6.** *RPN4* is upregulated by cytosolic mislocalization of secretory proteins. (A) Western blot of Kar2 and Pgk1 from wild-type (WT) and Δ*hac1* cells treated with 2 µg/ml tunicamycin for the times indicated. The asterisk indicates untranslocated ss-Kar2. (B) As in panel A, but with 5 µg/ml tunicamycin. (C) Western blot of Kar2 and Pgk1 from *sec65-1* cells grown at 25°C and shifted to the indicated temperatures for 90 min. The asterisk indicates untranslocated ss-Kar2. (D) *RPN4* mRNA levels in WT and *sec65-1* cells grown at 25°C and shifted to the indicated temperatures for 90 min.
*Figure 6 continued on next page*

*Figure 6 continued*

Data are normalized to WT cells at 25°C. Mean ±SEM, n = 3. (**E**) Flow cytometric measurement of GFP levels in WT and *sec65-1* cells harboring the *HAC1* splicing reporter. Cells grown at 25°C were shifted to the indicated temperatures or treated with 2 µg/ml tunicamycin (Tm) for 90 min. For each strain, data are normalized to 25°C. Mean ±SEM, n = 3.

DOI: https://doi.org/10.7554/eLife.43244.012

The following figure supplement is available for figure 6:

**Figure supplement 1.** *RPN4* mRNA levels in wild-type and *sec65-1* cells at different temperatures.

DOI: https://doi.org/10.7554/eLife.43244.013

misfolding induces the *RPN4* gene through the transcription factors Msn2/4, Yap1 and Hsf1. The resulting rise in Rpn4 protein levels reinforces activation of the Rpn4 regulon.

Rpn4 abundance is controlled by rapid proteasomal turnover (*Xie and Varshavsky, 2001*) and by *RPN4* gene activity. ER stress initially increases Rpn4 protein levels. Persistent ER stress, however, additionally raises *RPN4* mRNA levels. These observations suggest a biphasic Rpn4 response. First, when misfolded proteins occupy the proteasome, Rpn4 is spared from degradation and activates its target genes, but *RPN4* transcription remains unchanged. This scenario likely applies to early phases of ER stress and cases of mild stress. For instance, expression of the misfolded ER membrane protein Ste6* activates the Rpn4 regulon but not *RPN4* transcription (*Metzger and Michaelis, 2009*). If slowed Rpn4 degradation is insufficient to resolve the stress, *RPN4* transcription is upregulated in a second phase of the response, which provides another boost in Rpn4 target gene induction. Msn2/4, Yap1 and Hsf1 contribute to the stress-induced upregulation of *RPN4*, indicating that parallel pathways communicate protein misfolding to the *RPN4* promoter. This situation is distinct from *RPN4* induction after glucose starvation or heat shock, which requires Hsf1 but not Msn2/4 (*Hahn and Thiele, 2004a*). Interestingly, the *RPN4* promoter does not contain a canonical stress response element as binding site for Msn2/4 (*Martínez-Pastor et al., 1996*). Therefore, it is not clear whether Msn2/4 bind to the *RPN4* promoter directly, although high throughput data hint at this possibility (*Harbison et al., 2004*; *Huebert et al., 2012*). Furthermore, it remains to be determined exactly how cytosolic misfolding is sensed by Msn2/4, Hsf1 and Yap1. While the detection of misfolded cytosolic proteins by Hsf1 has been studied extensively, it is unknown how protein kinase A, which controls Msn2/4 and possibly also Hsf1 (*Verghese et al., 2012*), may monitor protein folding. Finally, it is unclear whether Yap1 is directly activated by misfolded proteins. Alternatively, its contribution to *RPN4* activation could reflect the fact that *YAP1* is an Rpn4 target gene (*Mannhaupt et al., 1999*) and may reinforce *RPN4* transcription as part of a positive feedback loop.

Rpn4 controls genes involved in many processes, including proteasome biogenesis, protein ubiquitination and DNA repair (*Mannhaupt et al., 1999*; *Jelinsky et al., 2000*). Hence, the Rpn4 regulon could increase ER stress resistance through a combination of functional outputs. However, it has been shown that the sensitivity of Δ*rpn4* cells to various stresses, including ER stress, results from their impaired induction of proteasome genes (*Wang et al., 2008*). Enhanced proteasome biogenesis is therefore likely to be critical for the promotion of ER stress resistance by Rpn4. Accordingly, overexpression of the Rpn4 target genes *YAP1* or *CDC48* increases ER stress resistance of UPR mutants much more weakly than *RPN4* overexpression. Therefore, the relevant functions of Rpn4 must extend beyond inducing Yap1-driven oxidative stress tolerance or Cdc48-dependent ERAD, and our data support the notion that enhancing degradation of cytosolically mislocalized secretory proteins is a major factor. Nevertheless, the relative contributions of different components of the Rpn4 regulon to ER stress resistance remain to be delineated.

An effective UPR requires sufficient proteasome capacity to remove misfolded proteins through ERAD. Yet, the UPR does not control genes encoding proteasome subunits (*Travers et al., 2000*). Furthermore, the only proteasome-related genes among the 102 genes we defined as core UPR targets were *PBA1* and *ADD66*, which encode proteasome assembly chaperones. Conversely, the Rpn4 regulon comprises most genes for proteasome subunits but only a few genes involved in ER protein folding, such as *PDI1* (*Mannhaupt et al., 1999*). Hence, the transcriptional programs activated by the UPR and Rpn4 are largely distinct. They are, however, functionally complementary and represent two cooperating modules. This functional complementarity explains the strong negative genetic interaction between *HAC1* and *RPN4*. Remarkably, ER stress activates a second signaling pathway that promotes proteasome biogenesis. The Slt2/Mpk1 MAP kinase, which augments ER stress

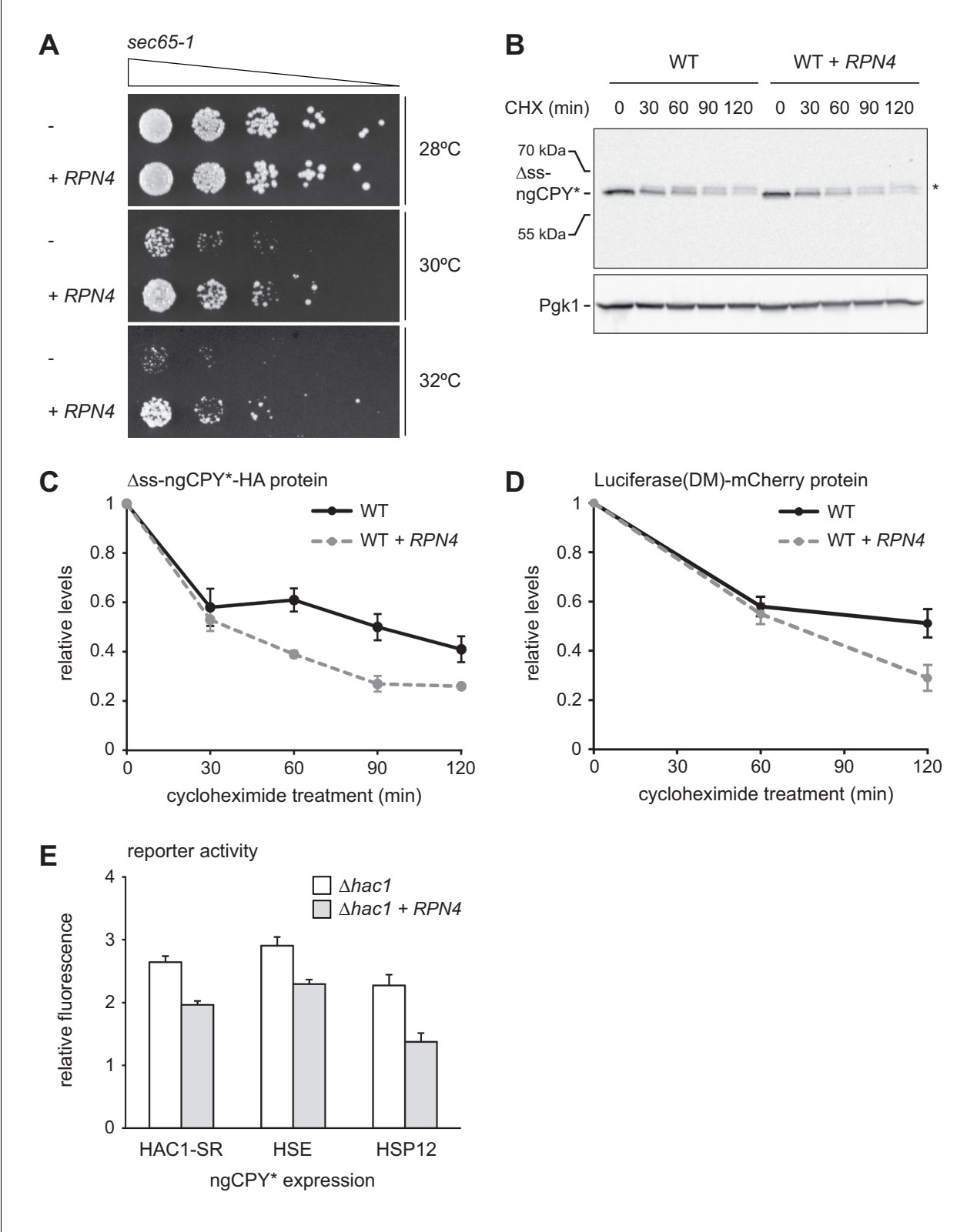

**Figure 7.** Rpn4 protects against cytosolic protein misfolding. (**A**) Growth assay on solid medium of *sec65-1* cells grown at different temperatures and overexpressing *RPN4* where indicated. (**B**) Western blot of HA and Pgk1 from cycloheximide-treated wild-type (WT) cells expressing Δss-ngCPY*-HA and additionally overexpressing *RPN4* where indicated. Expression of Δss-ngCPY*-HA was induced with 100 nM estradiol for 4 hr. The asterisk indicates a slower-migrating, post-translationally modified form of Δss-ngCPY*-HA. CHX, cycloheximide. (**C**) Quantification of Δss-ngCPY*-HA levels relative to

*Figure 7 continued on next page*

*Figure 7 continued*

Pgk1 from western blots as shown in panel B. For each strain, data are normalized to t = 0. Mean ±SEM, n = 3. (**D**) Luciferase(DM)-mCherry levels relative to Pgk1 and normalized to t = 0. Quantification is based on western blots of mCherry and Pgk1 from cycloheximide-treated cells expressing Luciferase(DM)-mCherry and additionally overexpressing *RPN4* where indicated. Mean ±SEM, n = 3. (**E**) Flow cytometric measurement of GFP levels in Δ*hac1* cells harboring the *HAC1* splicing reporter (HAC1-SR), HSE reporter or HSP12 reporter, expressing ngCPY* under the control of the estradiol-inducible *GAL* promoter system, and overexpressing *RPN4* where indicated. Expression of ngCPY* was induced with 100 nM estradiol for 5 hr. Data are normalized to cells not treated with estradiol. Mean ±SEM, n = 3.

DOI: https://doi.org/10.7554/eLife.43244.014

resistance, controls chaperones responsible for 19S regulatory particle assembly and increases the abundance of complete 26S proteasomes during stress (*Bonilla and Cunningham, 2003*; *Chen et al., 2005*; *Rousseau and Bertolotti, 2016*). Taken together, it is evident that proteasome biogenesis is an important UPR-independent process that enhances resistance to ER stress.

ER homeostasis and proteasome biogenesis are coupled also in higher eukaryotes, as shown by activation of the mammalian UPR upon proteasome inhibition (*Nishitoh et al., 2002*; *Obeng et al., 2006*). Furthermore, the mechanisms adjusting proteasome abundance in yeast and mammals share extensive similarities. Proteasome inhibition, which mimics proteasome overload, induces proteasome genes also in mammalian cells (*Mitsiades et al., 2002*; *Meiners et al., 2003*). This response is mediated by the transcription factor Nrf1 and the Slt2/Mpk1 homolog Erk5 (*Radhakrishnan et al., 2010*; *Steffen et al., 2010*; *Rousseau and Bertolotti, 2016*). Similar to Rpn4, Nrf1 is short-lived and activates many proteasome genes (*Radhakrishnan et al., 2010*; *Steffen et al., 2010*). Remarkably, Nrf1 is constitutively turned over by ERAD (*Steffen et al., 2010*). Hence, when ER stress overburdens ERAD, Nrf1 is stabilized and can promote proteasome biogenesis. Another intriguing parallel involves the mammalian proteasome assembly chaperones PAC1/2. As mentioned above, expression of their yeast counterparts *PBA1* and *ADD66* is induced by the UPR. PAC1/2 protein abundance also increases during ER stress, although this is achieved by iRhom1-mediated stabilization (*Lee et al., 2015*). Understanding the links between the UPR and proteasome biogenesis is relevant for human disease. Multiple myeloma cells suffer from chronic ER stress and are highly sensitive to proteasome inhibition, implying that proteasome capacity is limiting for survival of these cells. This insight has led to major improvements in the treatment of plasma cell cancer through the use of proteasome inhibitors (*Goldberg, 2012*).

A simple perspective on adaptive responses is that protein misfolding in a particular subcellular compartment triggers a dedicated program that enhances quality control specifically in the troubled organelle. Our work supports a more holistic view that emphasizes two additional concepts: functional modularity and cross-compartment coordination. Regulated proteasome biogenesis is a functional module that serves as part of the cellular response to ER stress. Interestingly, proteasome biogenesis is also activated by impaired mitochondrial protein import (*Wrobel et al., 2015*; *Wang and Chen, 2015*; *Boos et al., 2019*). It may therefore be a functional module that is commonly employed when stress necessitates clearance of mislocalized proteins from the cytosol. Autophagy and global attenuation of translation may be other such modules (*Gasch et al., 2000*). Furthermore, adaptive responses are not restricted to the compartment where stress initially arises. For example, as shown here, ER stress activates the UPR but also the Rpn4 regulon to safeguard against protein misfolding in the cytosol. Second, certain types of mitochondrial stress trigger the mitochondrial unfolded protein response but also activate the cytosolic heat shock response and promote proteasome biogenesis (*Ho et al., 2006*; *Wrobel et al., 2015*; *Kim et al., 2016*; *Boos et al., 2019*). Third, cytosolic folding stress activates the Hsf1-dependent heat shock response, which promotes protein folding in the cytosol but also controls the expression of major chaperones that function elsewhere, including Kar2 in the ER and Ssc1 in mitochondria (*Yamamoto et al., 2005*; *Hahn et al., 2004b*). Fourth, the UPR regulates genes that function in the entire secretory pathway (*Travers et al., 2000*). Many additional links between different organelle quality control systems exist in yeast and higher eukaryotes (*Higuchi-Sanabria et al., 2018*). A driving force for the evolution of these links may have been the need to prevent the spread of protein folding problems throughout the cell. Overall, it emerges that adaptive responses, although triggered by stress in one compartment, engage a combination of functional modules for comprehensive cell protection across compartment boundaries.

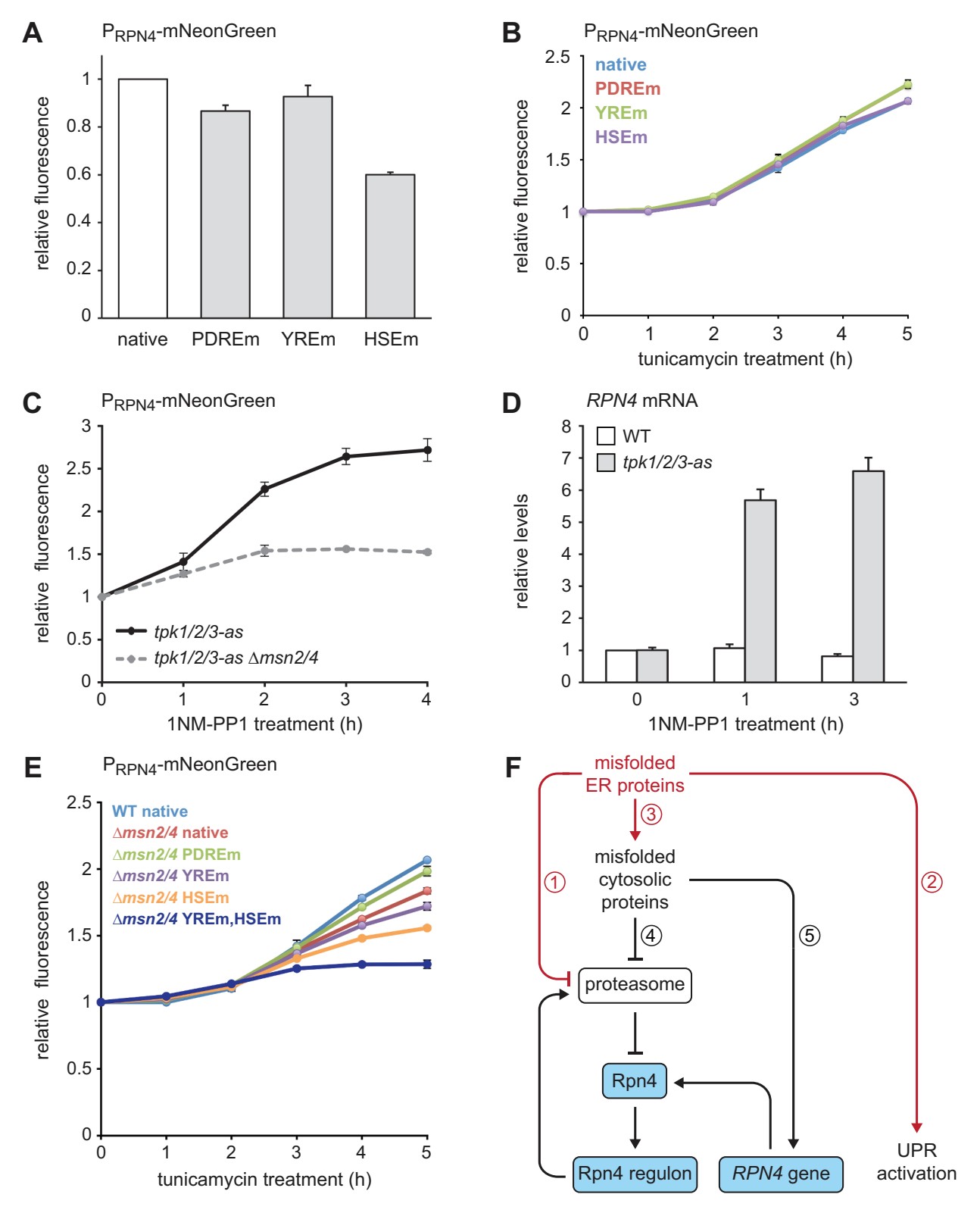

**Figure 8.** Multiple signaling pathways mediate *RPN4* induction by ER stress. (**A**) Flow cytometric measurement of the activity of *RPN4* reporter variants in untreated cells. Data are normalized to the reporter containing the native *RPN4* promoter. The other reporters contain mutations in the two Pdr1/3 response elements (PDREm), the Yap1 response element (YREm) or the heat shock element (HSEm). Mean ±SEM, n = 4. (**B**) As in panel A, but after treatment with 5 μg/ml tunicamycin for the times indicated. Mean ±SEM, n = 3. (**C**) Flow cytometric measurement of the activity of the native *RPN4*

*Figure 8 continued on next page*

*Figure 8 continued*

reporter in *tpk1/2/3-as* and *tpk1/2/3 Δmsn2/4* cells treated with the ATP analog 1NM-PP1 for the times indicated. The analog-sensitive *tpk1/2/3-as* alleles enable specific inhibition of protein kinase A with 1NM-PP1. Mean ±SEM, n = 3. (**D**) *RPN4* mRNA levels in wild-type (WT) and *tpk1/2/3-as* cells treated with 1NM-PP1 for the times indicated. Data are normalized to WT cells at t = 0. Mean ±SEM, n = 3. (**E**) Flow cytometric measurement of the activity of *RPN4* reporter variants in WT and *Δmsn2/4* cells treated with 5 µg/ml tunicamycin. Data are normalized to WT cells containing the reporter with the native *RPN4* promoter. Mean ±SEM, n = 3. (**F**) Model for the cooperation of Rpn4 and the UPR. ER protein misfolding causes increased flux through the ERAD pathway, which strains proteasome capacity and inhibits efficient protein degradation (1). In addition, the UPR is activated (2). Severe ER stress leads to translocation defects, causing mislocalization of secretory proteins to the cytosol, where they cannot fold properly (3). These cytosolic misfolded proteins further burden the proteasome (4). As a result, Rpn4 is stabilized, the Rpn4 regulon is activated and proteasome biogenesis is enhanced. If cytosolic misfolded proteins persist, the *RPN4* gene is induced (5), providing a second mechanism to increase Rpn4 abundance and augment proteasome biogenesis.

DOI: https://doi.org/10.7554/eLife.43244.015

# Materials and methods

## Key resources table

| Reagent type (species) or resource | Designation | Source or reference | Identifiers | Additional information |
|---|---|---|---|---|
| Strain (S. cerevisiae) | W303 | other | | see *Supplementary file 3* |
| Genetic reagent (E. coli) | *S. cerevisiae* genomic library in YEp13 E. coli | *Nasmyth and Tatchell, 1980* | ATCC 37323 | |
| Antibody | rat monoclonal anti-HA (clone 3F10) | Roche | Cat#11867423001; RRID: AB_390918 | (1:5000) |
| Antibody | rabbit polyclonal anti-mCherry | Biovision | Cat#5993; RRID: AB_1975001 | (1:5000) |
| Antibody | rabbit polyclonal anti-Kar2 | *Schuck et al., 2009* | | (1:50000) |
| Antibody | mouse monoclonal anti-Pgk1 (clone 22C5) | Abcam | Cat#Ab113687; RRID: AB_10861977 | (1:50000) |
| Antibody | goat anti-mouse HRP | Thermo Fisher Scientific Pierce | Cat#31432; RRID: AB_228302 | (1:10000) |
| Antibody | goat anti-rabbit HRP | Thermo Fisher Scientific Pierce | Cat#31462; RRID: AB_228338 | (1:10000) |
| Antibody | donkey anti-rat HRP | Jackson | Cat#712-035-153; RRID: AB_2340639 | (1:10000) |
| Recombinant DNA reagent | plasmids | other | DOI: 10.5061/dryad.n20d476 | see *Supplementary file 1* |
| Commercial assay or kit | NEBuilder HiFi DNA assembly master mix | New England Biolabs | Cat#E2611L | |
| Commercial assay or kit | BCA assay kit | Thermo Fisher Scientific Pierce | Cat#23225 | |
| Commercial assay or kit | ProtoScript II cDNA Synthesis kit | New England Biolabs | Cat#E6560L | |
| Commercial assay or kit | SensiFAST SYBR No-ROX kit | Bioline | Cat#BIO-98050 | |
| Chemical compound, drug | ß-estradiol | Sigma-Aldrich | Cat#E8875; CAS: 50-28-2 | |
| Chemical compound, drug | 1NM-PP1 | Merck | Cat#529581–1 MG; CAS: 221244-14-0 | |
| Chemical compound, drug | tunicamycin | Merck | Cat#654380–50 MG; CAS: 11089-65-9 | |
| Chemical compound, drug | cycloheximide | Sigma-Aldrich | Cat#C7698; CAS: 66-81-9 | |
| Chemical compound, drug | complete protease inhibitor | Roche | Cat#04693116001 | |
| Software, algorithm | Growthcurver | *Sprouffske and Wagner, 2016* | | |

*Continued on next page*

*Continued*

| Reagent type (species) or resource | Designation | Source or reference | Identifiers | Additional information |
|---|---|---|---|---|
| Software, algorithm | MaxQuant | *Cox et al., 2014* | | |
| Software, algorithm | Perseus | *Tyanova et al., 2016* | | |
| Other | CMAC stain | Thermo Fisher Scientific | Cat#C2110 | |
| Other | mass spectrometry proteomics data | this study | PRIDE database, PXD012867 | |

## Plasmids

Plasmids are listed in *Supplementary file 1*. Sequences of oligonucleotides for plasmid generation are given in *Supplementary file 2*. To generate plasmids for estradiol-inducible expression, $P_{GAL1}$-$T_{CYC}$ was amplified from pRS416-$P_{GAL1}$ (*Mumberg et al., 1994*) with primers up_SacI_GAL/CYC_KpnI_down and cloned into AgeI-linearized pRS306-$P_{ADH1}$-GEM or ApaI-linearized pNH605-$P_{ADH1}$-GEM (*Pincus et al., 2014*) by ligation with the NEBuilder HiFi DNA assembly master mix (New England Biolabs, Ipswitch, Massachusetts), yielding pRS306-$P_{ADH1}$-GEM-$P_{GAL1}$ and pNH605-$P_{ADH1}$-GEM-$P_{GAL1}$. Plasmids for expression of HA-tagged CPY* variants were subsequently generated by amplification of ngCPY*-HA or CPY*-HA from pRS315-$P_{CPY}$-ngCPY*-HA or pRS315-$P_{GAL1}$-CPY*-HA (*Spear and Ng, 2003*) with primers EDY*_F2/EDY*_R and ligation with SmaI-linearized pRS306-$P_{ADH1}$-GEM-$P_{GAL1}$, yielding pRS306-$P_{ADH1}$-GEM-$P_{GAL1}$-ngCPY*-HA and pRS306-$P_{ADH1}$-GEM-$P_{GAL1}$-CPY*-HA. Plasmids for expression of sfGFP-tagged CPY* variants were generated as follows: FLAG-sfGFP was amplified from pRS305-$P_{ADH1}$-Rtn1-FLAG-sfGFP (*Szoradi et al., 2018*) with primers FLAG-sfGFP_F/FLAG-sfGFP_R and ligated with SmaI-linearized pNH605-$P_{ADH1}$-GEM-$P_{GAL1}$, yielding pNH605-$P_{ADH1}$-GEM-$P_{GAL1}$-FLAG-sfGFP. Next, ngCPY*-HA was amplified from pRS315-$P_{CPY}$-ngCPY*-HA with primers EDY*_F2/EDY*_tag_R and ligated into pNH605-$P_{ADH1}$-GEM-$P_{GAL1}$-FLAG-sfGFP linearized with primers FLAG for 1/FLAG-open_R, yielding pNH605-$P_{ADH1}$-GEM-$P_{GAL1}$-ngCPY*-HA-sfGFP. Similarly, CPY*(N479Q)-HA was amplified from pRS305-$P_{CPY}$-CPY*(N479Q)-HA with primers EDY*_F2/EDY*_tag_R and ligated into pNH605-$P_{ADH1}$-GEM-$P_{GAL1}$-FLAG-sfGFP linearized with primers FLAG for 1/FLAG-open_R, yielding pNH605-$P_{ADH1}$-GEM-$P_{GAL1}$-CPY*(N479Q)-HA-sfGFP. To subclone genes of interest into YEp13, their coding regions together with upstream and downstream sequences were amplified from yeast genomic DNA with primers providing homologous ends (*Supplementary file 2*). The upstream and downstream sequences encompassed the entire sequence between the coding region of interest and the next upstream and downstream coding region, or at least 400 bp of upstream and 200 bp of downstream sequence. The resulting PCR products were recombined with BamHI-linearized YEp13 through gap repair cloning in yeast. To generate pRS306-$P_{ADH1}$-GEM-$P_{GAL1}$-Δss-ngCPY*-HA, pRS306-$P_{ADH1}$-GEM-$P_{GAL1}$-ngCPY*-HA was linearized by PCR with primers CPY deltaSS fw/CPY deltaSS rev and religated with the NEBuilder HiFi DNA assembly master mix, thereby eliminating the CPY signal sequence. To generate pRS304-HAC1-splicing-reporter, $P_{HAC1}$-Hac1-splicing reporter was amplified from pRS305-HAC1-splicing-reporter with primers pRS304-Eco-Hac1-SR fw/pRS304-Eco-Hac1-SR rev II and ligated with EcoRI-linearized pRS304 (*Sikorski and Hieter, 1989*) using the NEBuilder HiFi DNA assembly master mix. To generate pNH604-$P_{HSP12}$-GFP, $P_{HSP12}$-GFP was excised from pNH605-$P_{HSP12}$-GFP with PspOMI/SacII and cloned into the PspOMI/SacII site of pNH604-4x$P_{HSE}$-YFP. To generate pNH605-$P_{RPN4}$-mNeon-Green, mNeonGreen was amplified from pFA6a-mNeonGreen-kanMX4 with primers Xho-neon/neon-BamHI and cloned between the XhoI and BamHI sites of pNH605-$P_{RPN4}$-YFP. Similarly, pNH605-$P_{RPN4}$-(HSEm)-mNeonGreen and pNH605-$P_{RPN4}$-(PDREm)-mNeonGreen were generated by replacing YFP in pNH605-$P_{RPN4}$-(HSEm)-YFP and pNH605-$P_{RPN4}$-(PDREm)-YFP with mNeonGreen. To generate pNH605-$P_{RPN4}$-(YREm)-mNeonGreen and pNH605-$P_{RPN4}$-(YREm,HSEm)-mNeonGreen, pNH605-$P_{RPN4}$-mNeonGreen or pNH605-$P_{RPN4}$-(HSEm)-mNeonGreen were linearized with primers NheI-YRE fw/NheI-YRE rev, digested with NheI and religated, thereby mutating the YRE to an NheI site.

## Yeast strain generation

Strains used in this study are listed in *Supplementary file 3*. Unless indicated otherwise, strains were derived from W303 mating type a (strain SSY122). Gene tagging, gene deletion and promoter replacement was done with PCR products with homologous ends (*Longtine et al., 1998*; *Janke et al., 2004*). For irreversible single-copy genomic integration of CPY* expression plasmids, the $P_{ADH1}$-GEM-$P_{GAL1}$ expression cassette was amplified with primers knock-in URA3 fw/knock-in URA3 rev (*Supplementary file 2*) and integrated into the *URA3* locus. For integration of *TRP1*-marked pRS304-HAC1-SR into the *LEU2* locus, the HAC1-splicing-reporter-TRP1 expression cassette was amplified with primers knock-in LEU2 fw/knock-in LEU2 rev (*Supplementary file 2*). Other integrative plasmids were linearized by restriction digest before transformation.

## Growth conditions

Strains were cultured at 30°C in SCD medium consisting of 0.7% yeast nitrogen base, 2% glucose and amino acids, lacking leucine where appropriate to maintain plasmid selection. *Sec65-1* cells were grown in the same medium at 25°C. For steady state analyses, cultures were grown to saturation, diluted and grown for at least 9 hr so that they reached mid log phase ($OD_{600}$ = 0.5–1). For induction of CPY* expression, exponentially growing cells were diluted to early log phase ($OD_{600}$ = 0.1–0.5) and treated with 50 nM β-estradiol (Sigma-Aldrich, St. Louis, Missouri) for 4 hr, unless indicated otherwise. For cycloheximide chase experiments, cells in mid log phase were treated with 50 µg/ml cycloheximide (Sigma-Aldrich). For tunicamycin treatment, exponentially growing cells were diluted to early log phase and treated with 2 or 5 µg/ml tunicamycin (Merck, Darmstadt, Germany) as indicated. For 1NM-PP1 treatment, exponentially growing cells were diluted to early log phase and treated with 3 µM 1NM-PP1 (Merck). For temperature shift experiments, cells grown to mid log phase at 25°C were diluted to $OD_{600}$ = 0.2 and incubated at the indicated temperatures for 90 min.

## *HAC1* splicing, *HSE*, *HSP12* and *RPN4* reporter assays

To measure UPR activity, a *HAC1* splicing reporter was used that translates Ire1 activity into the production of GFP (*Pincus et al., 2010*). To measure induction of the UPR, cells harboring this reporter were grown to mid log phase in 1 ml medium in 96 deep-well plates. Cells were diluted to early log phase, treated with estradiol or tunicamycin as described above, 100 µl aliquots were removed at each time point and GFP fluorescence after excitation with a 488 nm laser was measured with a FACS Canto flow cytometer (BD Biosciences, Franklin Lakes, New Jersey) equipped with a high-throughput sampler. In parallel, autofluorescence was determined with identically grown isogenic control strains not harboring the splicing reporter. Mean cellular GFP fluorescence was corrected for autofluorescence and normalized to the GFP fluorescence of untreated cells. To measure steady state UPR activity in different strains, cells expressing the *HAC1* splicing reporter and cytosolic BFP under the control of the constitutive *GPD* promoter were grown to mid log phase as above and GFP and BFP fluorescence were measured after excitation with 488 nm or 405 nm lasers. GFP fluorescence was corrected for autofluorescence and divided by BFP fluorescence to account for differences in protein translation capacity. Data were expressed relative to the GFP/BFP fluorescence ratio in wild-type cells.

To measure Hsf1 and Msn2/4 activity, cells harboring the 4xHSE-YFP or the HSP12-GFP reporter and expressing ngCPY* under the control of the estradiol-inducible *GAL* promoter system were grown to mid log phase as above and diluted to early log phase. Cells were either left untreated or were treated with 100 nM estradiol for 5 hr, 100 µl aliquots were removed and GFP fluorescence was measured. GFP fluorescence was corrected for autofluorescence and corrected fluorescence of treated cells was normalized to that of corresponding untreated cells.

To measure Rpn4 activity, cells harboring an *RPN4* reporter and expressing cytosolic BFP under the control of the constitutive *GPD* promoter were grown to mid log phase as above. To measure steady state *RPN4* activity, 100 µl aliquots were removed and mNeonGreen and BFP fluorescence was measured after excitation with 488 nm or 405 nm lasers as above. Mean cellular mNeonGreen fluorescence was corrected for autofluorescence as above and divided by BFP fluorescence. Data were expressed relative to the mNeonGreen/BFP fluorescence ratio in cells harboring the wild-type *RPN4* reporter. To measure induction of *RPN4*, cells were grown to mid log phase as above, diluted

to early log phase and either left untreated or were treated with 5 µg/ml tunicamycin or 3 µM 1NM-PP1. At each time point, 100 µl aliquots were removed and mNeonGreen and BFP fluorescence was determined. For each time point, autofluorescence-corrected mNeonGreen/BFP ratios were calculated as above and ratios in treated cells were normalized to those in corresponding untreated cells.

## Western blotting

Cell lysis and western blotting was done as described (*Szoradi et al., 2018*). In brief, cells were disrupted by bead beating, proteins were solubilized with SDS, protein determination was carried out with the BCA assay kit (Thermo Fisher Scientific Pierce, Waltham, Massachusetts), equal amounts of protein were resolved by SDS-PAGE and transferred onto nitrocellulose membranes. Membranes were probed with primary and HRP-coupled secondary antibodies, developed with homemade ECL, and chemiluminescence was detected with an ImageQuant LAS 4000 imaging system (GE Healthcare, Chalfont St Giles, UK). Images were quantified with ImageJ and processed with Adobe Photoshop. Primary antibodies were rat anti-HA 3F10 (Roche, Basel, Switzerland), rabbit anti-Kar2 (Peter Walter, UCSF), rabbit anti-mCherry (Biovision, Milpitas, California) and mouse anti-Pgk1 22C5 (Abcam, Cambridge, UK).

## Light microscopy

CPY* expression was induced with 25 nM estradiol for 4 hr. Ten µM CMAC (Thermo Fisher Scientific, Waltham, Massachusetts) was added during the last 2 hr of induction to stain the vacuole and cells were imaged live at room temperature. Images were acquired with a DMi8 inverted microscope (Leica, Wetzlar, Germany) equipped with a CSU-X1 spinning-disk confocal scanning unit (Yokogawa, Musashino, Japan), a ORCA-Flash 4.0 LT camera (Hamamatsu, Hamamatsu, Japan) and a HC PL APO 100x/1.4 NA CS2 oil objective lens (Leica). Background subtraction with a rolling ball algorithm was performed in ImageJ and images were processed in Adobe Photoshop.

## Growth assays

Growth assays on agar plates and in liquid medium were done as described (*Schuck et al., 2009*; *Szoradi et al., 2018*). For growth assays on agar plates, dilution series with fivefold dilution steps were used. For quantification of growth in liquid medium, the cell density in arbitrary units was plotted against time and the area under the curve was calculated with the R package Growthcurver (*Sprouffske and Wagner, 2016*). Data were normalized to the wild-type control and expressed as a growth index, which was set to one for wild-type cells.

## Viability assay

Exponentially growing cells were diluted to $OD_{600}$ = 0.05 and grown in the presence of different concentrations of β-estradiol for 24 hr. Cultures were diluted to equal cell densities as judged by $OD_{600}$ measurements, equal numbers of cells were plated on solid YPD medium (1% yeast extract, 2% peptone, 2% glucose) and grown for 48 hr. To determine cell viability, the number of colony-forming units of estradiol-treated samples was normalized that of the mock-treated sample.

## Genetic screen

Strain SSY1341 was transformed with a yeast genomic library in the YEp13 multicopy vector (*Nasmyth and Tatchell, 1980*; available from the American Type Culture Collection as ATCC 37323; kindly provided by Michael Knop, ZMBH). Transformants were plated onto SCD-Leu plates at approximately 200 colony-forming units per plate and grown at 30°C. After 26 hr, colonies were replicated onto SCD-Leu plates containing 50, 75 or 100 nM estradiol and grown for up to 36 hr. Colonies that clearly grew better than the general background were restreaked onto SCD-Leu plates and replicated onto SCD-inositol to identify transformants that grew due to re-expression of *HAC1*. For confirmation, plasmids were retrieved from six transformants that grew in the absence of inositol and sequenced with primers YEp13 fw/YEp13 rev (*Supplementary file 2*). All contained *HAC1*. Transformants that failed to grow without inositol and hence lacked *HAC1* were re-tested by growth assays on SCD-Leu plates containing 50 nM estradiol. Suppressing plasmids were retrieved from well-growing transformants and their inserts were sequenced. Inserts contained between one and six genes. To determine which genes were responsible for suppression, candidates were individually

subcloned into YEp13 from genomic DNA from SSY122 and tested for growth on SCD-Leu plates containing 50 nM estradiol. Genes identified only once were discarded, with the exception of *PDR1* and *YAP1*.

## Quantitative real-time PCR

Isolation of mRNA, cDNA synthesis and quantitative real-time PCR were done as described (*Szoradi et al., 2018*). In brief, RNA from 5 ODs of cells was extracted with hot phenol, precipitated with ethanol and resuspended in 30 µl $H_2O$. Synthesis of cDNA was done from 0.5 µg total RNA with the Protoscript II kit (New England Biolabs) using $d(T)_{18}$ primers. PCRs containing 5 ng template DNA and 250 nM each of forward and reverse primers were prepared using the SensiFAST SYBR No-ROX kit (Bioline, Luckenwalde, Germany). Primer sequences are listed in *Supplementary file 2*. PCRs were run in triplicate on a LightCylcer II 480 (Roche) with an annealing temperature of 60°C and an extension time of 20 s. The *TAF10* mRNA served as internal standard to determine relative mRNA levels of *KAR2*, *SIL1* or *RPN4*. Data analysis was performed with the LightCylcer II 480 software using the $2^{nd}$ derivative maximum method to determine Cp (crossing point) values.

## Proteomics

Wild-type (SSY122) and Δ*rpn4* (SSY784) cells were grown to mid-log phase in SCD medium. Cultures were diluted to OD 0.4, and 5 ODs of cells were harvested as untreated samples. Tunicamycin was added to the remainder of each culture at a final concentration of 5 µg/ml, cells were grown for 4 hr, and 5 ODs of cells were harvested as treated samples. Cells were collected by centrifugation and snap frozen in liquid nitrogen. Samples from four independent experiments were used for proteomic analysis. Cells were resuspended in ice-cold lysis buffer (50 mM Tris, pH 7.5, 0.5 mM EDTA, 1x Roche complete protease inhibitors) and disrupted by bead beating. Proteins were solubilized by adding SDS to a final concentration of 1.8% (w/v) and heating to 95°C for 10 min. Protein concentrations were determined with a BCA assay kit (Thermo Fisher Scientific). Per sample, 200 µg protein was precipitated with acetone and resuspended in digestion buffer (8 M Urea, 50 mM Tris pH 8.5, 1 mM DTT) to a concentration of 4 mg/ml. 40 µg protein was alkylated using 5 mM iodoacetamide for 1 hr, digested with LysC (enzyme to protein = 1:50 (w/w)) for 5.5 hr, diluted to 2 M Urea, and digested with Trypsin (1:50 (w/w)) for another 13 hr at room temperature. Digested peptides were acidified to 1% (v/v) trifluoroacetic acid, cleared of precipitates by brief centrifugation, desalted via SDB-RPS cleanup, and analyzed on a Q Exactive HF-X Hybrid Quadrupole-Orbitrap mass spectrometer (Thermo Fisher Scientific), essentially as described (*Itzhak et al., 2016*). Raw files were processed using MaxQuant (*Cox and Mann, 2008*) version 1.6, using the MaxLFQ algorithm (*Cox et al., 2014*) for label free-quantification. Downstream analysis was performed in Perseus version 1.6 (*Tyanova et al., 2016*). LFQ intensities based on fewer than 2 MS/MS counts were removed from the dataset. The remaining intensities were log-transformed. For the pairwise analyses in *Figures 4D* and *5F*, further quality filtering was applied. Only proteins that were quantified in all four replicates of at least one of the compared conditions were retained. Missing values were then imputed from a normal distribution (width 0.3 SDs, down-shifted by 1.8 SDs). For statistical analysis, a two-sided student's t-test with permutation based false discovery rate control (FDR = 1%) and an S0 parameter of 0.2 was performed. The category annotation enrichment was calculated with the 1D annotation enrichment tool in Perseus (*Tyanova et al., 2016*), using mean $log_2$ expression differences. For the analysis in *Figure 5—figure supplement 2*, only proteins annotated as 'proteasome subunit' and with at least three quantified intensities in each condition were included. For each protein, the median value of the untreated wild-type sample was subtracted from the median intensities in all conditions, to achieve normalization. The resulting values correspond to $log_2$ fold changes relative to the untreated wild-type (which has a value of 0). The mean and standard error of the mean were plotted. Proteasome subunits were defined as the 33 structural proteins of the 20S core particle and the 19S regulatory particle (*Supplementary file 4*). All were detected consistently, except for Sem1. Core UPR target genes were defined as those identified as UPR-regulated by both *Travers et al. (2000)* and *Pincus et al. (2014)* (*Supplementary file 5*). Of the corresponding 102 proteins, 69 were detected in total, and 50 and 58 passed the quality filters for the analyses in *Figures 4D* and *5F*, respectively. See *Figure 4—source data 1* for assignment of proteins to other functional groups.

## Experimental design

Control strains were isogenic to the experimental strains except for the relevant genetic modifications. For experiments with *sec65-1* cells, strain SSY002 (W303 mating type alpha) rather than SSY122 was used as a control because it has the same mating type and also is an *ade2* mutant. At least three biological replicates were done for experiments with quantitative read-outs to enable assessment of the variation between replicates. Exceptions were the experiments in *Figure 1E*, *Figure 1F*, *Figure 1—figure supplement 1D* and *Figure 6—figure supplement 1*, which were done only once. Repetitions were considered biological replicates if they were initiated from independently inoculated pre-cultures of the yeast strains used and were performed on different days. For each experiment, the number of biological replicates (n), the mean and the standard error of the mean (SEM) are reported in the figure legends.

## Data availability

The mass spectrometry proteomics data associated with *Figure 4D*, *Figure 5F* and *Figure 5—figure supplement 2* have been deposited to the ProteomeXchange Consortium via the PRIDE partner repository with the dataset identifier PXD012867.

## Acknowledgements

We thank Felix Boos, Michael Knop, David Pincus and Peter Walter for reagents, the Flow Cytometry and FACS facility at the ZMBH for assistance, Dorottya Polos, Anja Klemmer and Jan Grosser for contributions to early stages of this study, and David Pincus, Davis Ng, Felix Boos, Anne-Lore Schlaitz and all members of the Schuck lab for comments on the manuscript. This work was funded by a PhD fellowship from the Heidelberg Biosciences International Graduate School (HBIGS) to RMS, the Max Planck Society for the Advancement of Sciences, and grants EXC 81 and MA 1764/2-1 from the Deutsche Forschungsgemeinschaft (DFG).

## Additional information

### Funding

| Funder | Grant reference number | Author |
|--------|------------------------|--------|
| Heidelberg Biosciences International Graduate School | Graduate Student Fellowship | Rolf M Schmidt |
| Deutsche Forschungsgemeinschaft | EXC 81 | Rolf M Schmidt Sebastian Schuck |
| Max-Planck-Gesellschaft | | Julia P Schessner Georg HH Borner |
| Deutsche Forschungsgemeinschaft | MA 1764/2-1 | Georg HH Borner |

The funders had no role in study design, data collection and interpretation, or the decision to submit the work for publication.

### Author contributions

Rolf M Schmidt, Conceptualization, Investigation, Writing—review and editing; Julia P Schessner, Formal analysis, Investigation, Writing—review and editing; Georg HH Borner, Formal analysis, Writing—review and editing; Sebastian Schuck, Conceptualization, Investigation, Writing—original draft, Writing—review and editing

### Author ORCIDs

Rolf M Schmidt http://orcid.org/0000-0002-1263-2406
Georg HH Borner http://orcid.org/0000-0002-3166-3435
Sebastian Schuck http://orcid.org/0000-0002-6388-0661

Decision letter and Author response
Decision letter https://doi.org/10.7554/eLife.43244.027
Author response https://doi.org/10.7554/eLife.43244.028

## Additional files

### Supplementary files

• Supplementary file 1. Plasmids used in this study.
DOI: https://doi.org/10.7554/eLife.43244.016

• Supplementary file 2. Oligonucleotides used in this study.
DOI: https://doi.org/10.7554/eLife.43244.017

• Supplementary file 3. Yeast strains used in this study.
DOI: https://doi.org/10.7554/eLife.43244.018

• Supplementary file 4. Proteasome subunits.
DOI: https://doi.org/10.7554/eLife.43244.019

• Supplementary file 5. Core UPR target genes.
DOI: https://doi.org/10.7554/eLife.43244.020

• Transparent reporting form
DOI: https://doi.org/10.7554/eLife.43244.021

### Data availability

The mass spectrometry proteomics data associated with Figure 4D, Figure 5F and Figure 5 - figure supplement 2 have been deposited to the ProteomeXchange Consortium via the PRIDE partner repository with the dataset identifier PXD012867.

The following datasets were generated:

| Author(s) | Year | Dataset title | Dataset URL | Database and Identifier |
|---|---|---|---|---|
| Rolf M Schmidt, Julia P Schessner, Georg HH Borner, Sebastian Schuck | 2019 | Data from: The proteasome biogenesis regulator Rpn4 cooperates with the unfolded protein response to promote ER stress resistance | https://dx.doi.org/10.5061/dryad.n20d476 | Dryad Digital Repository, 10.5061/dryad.n20d476 |
| Schmidt RM, Schessner JP | 2019 | The proteasome biogenesis regulator Rpn4 cooperates with the unfolded protein response to promote ER stress resistance | http://www.ebi.ac.uk/pride/archive/projects/PXD012867 | PRIDE database, PXD012867 |

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
