## [Decision Letter]

Thank you for submitting your article "The proteasome biogenesis regulator Rpn4 cooperates with the unfolded protein response to promote ER stress resistance" for consideration by *eLife*. Your article has been reviewed by three peer reviewers, one of whom is a member of our Board of Reviewing Editors, and the evaluation has been overseen by Anna Akhmanova as the Senior Editor. The following individual involved in the review of your submission has agreed to reveal his identity: David Pincus (Reviewer #3).

The reviewers have discussed the reviews with one another and the Reviewing Editor has drafted this decision to help you prepare a revised submission.

In this manuscript, the authors use a titratable ER stress inducing system in yeast to identify specific stress-responsive pathways that coordinate with the UPR to protect cells against ER stress. In a suppressor screen, they show that overexpression of *RPN4* (and many other genes known to regulate *RPN4*) rescues growth defects in yeast deficient in the UPR-associated transcription factor treated with different types of ER stress. *RPN4* is the yeast transcription factor primarily responsible for regulating proteasome activity. They go on to show that *rpn-4* and *hac-1* synthetically influence yeast growth in the absence and presence of stress, suggesting cooperative protective activities for these transcription factors. In response to ER stress, the authors propose that impaired ERAD and the accumulation of mislocalized ER proteins in the cytosol challenge proteasome activity resulting in *RPN4* activation. Initially, *RPN4* is post-transcriptionally regulated, likely through impaired proteasomal degradation, but at later points, *rpn-4* is transcriptionally induced in a process dependent on multiple stress-responsive transcription factors including *HSF1* and *YAP*. Ultimately, the authors propose that *RPN4* and *HAC1* represent two complementary signaling mechanisms involved in coordinating the cellular response in response to ER stress.

Overall, this is an interesting manuscript addressing important questions related to the coordination of stress-pathways in response to diverse insults. The experiments are generally well performed and the conclusions are supported by the experimental data. However, we have identified three points that we feel must be addressed in revision and also ask you to consider (at your discretion) other points that came up in review, provided they can may be addressed in short order.

Essential revisions:

1) Drawing on point 1 of reviewer 1 and 2 we note that the genetic link to Rpn4 is a key conclusion of the paper. As you speculate that some of the other hit in the high-dose suppressor screen may encode genes that operate upstream of Rpn4, we think it is reasonable to ask that you apply genetic tests to examine this point.

2) Drawing on point 2 of reviewers 1 and 2, and also referred to by reviewer 3, we feel that the question whether Rpn4 upregulation impacts favourably on the well-being of ER stressed cells by helping restore protein folding homeostasis in the ER or by some other mechanism is a key element of the paper. Therefore, we ask that you try to address this point further. Whether you do so by measuring the level of *IRE1* activity (with a splicing reporter) or by directly measuring the buildup of unfolded proteins, or by some other more clever means, will be up to you.

3) Drawing on point 4 of reviewers 2 and the comments of reviewer 3, we ask that you extend the analysis beyond Rpn4 and examine the effects of unmitigated ER stress on expression of proteasome subunits and other genes in the proteasome biogenesis sphere.

*Reviewer #1:*

This paper presents a thorough analysis of the interactions between protein quality control pathways in the yeast ER and cytosol. The key insights have been gleaned from application of an unbiased technique: a genomic high dose suppressor library. This genome wide screen revealed the capacity for enhanced expression of Rpn4 (a transcription factor concerned with activation of genes important for proteasomal biogenesis) and potentially allied genes, to buffer the consequence of deficiencies in the UPR in ER-stressed yeast.

The findings here give rise to some interesting, unanticipated conclusions. Notably that the aforementioned buffering occurs despite the fact that the transcriptional control of genes that enhance the protein folding capacity of the ER and the genes involved in proteasome biogenesis occurs by parallel pathways.

There two points of critique that I feel should be addressed:

The authors present a strong case that in the unbuffered state (e.g. tunicamycin-treated ∆*hac1* yeast) activation of Rnp4 arises by the presence in the cytosol of proteins that would otherwise find their way into the ER. This begs the question if the buffering of the growth defect arising from Rnp4 over-expression reflects predominantly the role of the proteasome in alleviating this cytosolic problem, or if the presumptive enhanced proteasomal function of Rpn4(+) cells also reflects a cross compartment effect, whereby the Rpn4(+); E2-treated ngCPY*; ∆*hac1* cells also experience lower levels of ER stress; lower than the comparison group in which Rpn4 is not over-expressed? It may be possible to gage the level of unfolded protein stress in the ER by monitoring the state of Ire1p activity. I'm not sure if it is possible to monitor endogenous Ire1p phosphorylation in budding yeast, but perhaps this question can be answered by introducing an inert Hac1 splicing reporter into the experimental system.

The authors speculate on the possibility that buffering observed by Pdr1 and Ssz1 over-expression may be channelled through their role in raising the levels of Rpn4, thus providing a unifying mechanism of action for the top hits in this screen. Can this notion be tested? Is the buffering by Pdr1(+) and Ssz1(+) of E2-treated ngCPY*; ∆*hac1* cells eliminated by ∆Rpn4?

*Reviewer #2:*

In this manuscript, the authors use a titratable ER stress inducing system in yeast to identify specific stress-responsive pathways that coordinate with the UPR to protect cells against ER stress. In a suppressor screen, they show that overexpression of *RPN4* (and many other genes known to regulate *RPN4*) rescues growth defects in yeast deficient in the UPR-associated transcription factor treated with different types of ER stress. *RPN4* is the yeast transcription primarily responsible for regulating proteasome activity. They go on to show that *rpn-4* and *hac-1* synthetically influence yeast growth in the absence and presence of stress, suggesting cooperative protective activities for these transcription factors. In response to ER stress, the authors propose that impaired ERAD and the accumulation of mislocalized ER proteins in the cytosol challenge proteasome activity resulting in RPN4 activation. Initially, *RPN4* is post-transcriptionally regulated, likely through impaired proteasomal degradation, but at later points, *rpn*-4 is transcriptionally induced in a process dependent on multiple stress-responsive transcription factors including *HSF1* and *YAP*. Ultimately, the authors propose that *RPN4* and *HAC1* represent two complementary signaling mechanisms involved in coordinating the cellular response in response to ER stress.

Overall, this is an interesting manuscript addressing important questions related to the coordination of stress-pathways in response to diverse insults. The experiments are generally well performed and the conclusions are supported by the experiments. However, I have a few specific comments primarily pertaining the relationship between *RPN4* activation and specific biologic functions during ER stress that should be addressed prior to publication. I outline these comments below.

1) The authors suggest that other genes identified in their screen (e.g., *PDR1, SSZ1*, and *YAP1*) promote cellular growth of *hac1*-deficient cells in the presence of ER stress through activation of *RPN4*. If this is the case, one would predict that overexpression of these genes would not rescue growth in *rpn4*-deficient yeast incubated with increasing amounts of ER stress. This would provide further evidence that these genes function in the screen through the upregulation of *RPN4*, as suggested. Alternatively, this could function through alternative mechanisms. Either way, it would be interesting to define the specific dependence of gene rescue from these different hits on *RPN-4*.

2) The authors suggest that the increased UPR afforded by ER stress in *rpn-4* deficient yeast results from the ER buildup of misfolded ERAD substrates. This is probably true, but it would be good to see the increased ER accumulation of these proteins to confirm this model.

3) The authors suggest that accumulation of ER proteins through either increased ERAD or impaired translocation is responsible for the increased *RPN4* activity observed during ER stress. They do a nice job of testing the impaired translocation using the *sec65* mutant. However, it would be interesting to define the importance of ERAD in this process using *hrd-1* deficient yeast. If ERAD is required, one would expect that overexpression of *RPN4* should not rescue growth under these conditions. It would be nice to complete the study by testing this alternative mechanism of increased cytosolic protein load afforded by increased ERAD.

4) The idea is that increases in *RPN4* coordinate with the UPR by upregulating proteasome biogenesis (i.e., increasing proteasome capacity). However, increased expression of proteasome subunits and measures of proteasome activity are not provided (although they do show modest increases in the degradation of cytosolic proteins in cells following *RPN4* overexpression). If *RPN4* is protective during ER stress through increased proteasome biogenesis, shouldn't there be a measurable increase in proteasome capacity? Also, shouldn't there be an increase in proteasome subunit expression? This type of information should be provided to support the overall model proposed in panel 7F.

5) Along the same lines as #3, does ER stress induce proteasome biogenesis in mammalian cells? There are lots of transcriptional profiles available, and it would be interesting to see if this model is conserved in higher eukaryotes. If not, are there other mechanisms involved that perform a similar function such as the iRHom regulation during ER stress in flies reported by Lee et al., 2015?

6) The authors show that they can suppress the transcriptional upregulation of *RPN4* by deletion of *MSN2/4* and removal of the YRE and HSE from the *rpn-4* promoter. Can they use this approach to define the relative importance of posttranscriptional and transcriptional regulation of *RPN4* during ER stress? It would be interesting to define the specific dependence of yeast protection against ER stress on these two aspects of *RPN4* regulation.

*Reviewer #3:*

This paper by Schmidt and Schuck is a valuable contribution to our understanding of cellular stress response signaling. The major conclusion is that the accumulation of unfolded proteins in the lumen of the ER does not merely activate the UPR(er), but rather triggers a chain of events that results in mistargeting of ER clients to the cytosol where they trigger a secondary set of proteostasis pathways. These pathways converge on the transcription of *RPN4*, which encodes the master transcriptional regulator of proteasome biogenesis, thereby promoting protein degradation. The experiments are well executed and clearly presented in the text and figures, and the data are convincing.

To arrive at their conclusions, the authors develop a titratable induction system in yeast that allows them to tune the level of unfolded proteins in the ER, and they couple it to fluorescent reporters that allow them to measure activity in the UPR and other stress response pathways. In my opinion, this system now represents the gold standard for physiological/homeostatic probing of the UPR and should be refactored and ported to mammalian cells. They also nicely connected the dots between ER stress and overload to cytosolic mistargeting using a Kar2-SS cleavage assay and a *sec65*-ts mutant.

The only aspect of the paper that is wanting is the end, where the authors bash the *RPN4* promoter and use analog sensitive PKA and *msn2/4*∆ strains to attempt to define the signals that activate *RPN4*. The approach here is a bit muddled and as such the conclusion is not clear. To bring this home, I would recommend doing a clean necessary/sufficient scan of the TF binding sites in the *RPN4* promoter. Start with 1kb upstream of *RPN4*: there are 6 STREs, 2 PDREs, 1 YRE and 1 HSE (I am happy to send my analysis of where the STREs are, as apparently the authors could not find them). In addition to the PDRE, YRE and HSE mutants the authors have, order a GBlock lacking the STREs and do the RT-qPCR. This will complete the necessity set. For sufficiency, just order 4 Gblocks that have only the 6 STREs, 2PDREs, the YRE or the HSE. Then add Tm and induced ngCPY* and measure *RPN4* expression and growth in this set. The PKA/*msn2/4* mutants, while clever, don't make the point as clearly.

---

## [Author Response]

Essential revisions:1) Drawing on point 1 of reviewer 1 and 2 we note that the genetic link to Rpn4 is a key conclusion of the paper. As you speculate that some of the other hit in the high-dose suppressor screen may encode genes that operate upstream of Rpn4, we think it is reasonable to ask that you apply genetic tests to examine this point.

We clarified this point by generating strains in which the *RPN4* gene was placed under the control of the constitutive *CYC1* promoter and thus uncoupled from its normal regulation. Overexpression of *PDR1, SSZ1* or *YAP1* still promoted ER stress resistance in this background. Therefore, Pdr1, Ssz1 and Yap1 can act through mechanisms unrelated to *RPN4* gene activation (subsection “A screen for genes promoting ER stress resistance in UPR mutants”, second paragraph and new Figure 3A). We also tried to do similar experiments in *∆hac1 ∆rpn4* cells but their extremely poor growth even in the absence of exogenous ER stressors (see Figure 3B and 3C) rendered these attempts uninformative.

2) Drawing on point 2 of reviewers 1 and 2, and also referred to by reviewer 3, we feel that the question whether Rpn4 upregulation impacts favourably on the well-being of ER stressed cells by helping restore protein folding homeostasis in the ER or by some other mechanism is a key element of the paper. Therefore, we ask that you try to address this point further. Whether you do so by measuring the level of IRE1 activity (with a splicing reporter) or by directly measuring the build-up of unfolded proteins, or by some other more clever means, will be up to you.

Using reporters for the activity of Ire1, Hsf1 and the stress response transcription factors Msn2/4, we now show that *RPN4* overexpression can improve protein folding homeostasis in both the ER and the cytosol of ER-stressed cells (subsection “Rpn4 is upregulated by and protects against mislocalized secretory proteins”, last paragraph and new Figure 7E). This finding supports the proposed cross-compartment response to ER stress.

3) Drawing on point 4 of reviewers 2 and the comments of reviewer 3, we ask that you extend the analysis beyond Rpn4 and examine the effects of unmitigated ER stress on expression of proteasome subunits and other genes in the proteasome biogenesis sphere.

To address this point, we performed proteomic analyses of untreated and tunicamycin-treated wild-type and *∆rpn4* cells. As anticipated from the activation of Rpn4 by ER stress, the expression levels of proteasome subunits and proteasome assembly chaperones rose in tunicamycin-treated wild-type cells (subsection “Rpn4 and the UPR cooperate to counteract ER stress”, last paragraph and new Figure 5F). A comparison of wild-type and *∆rpn4* cells at steady state showed a reduction in proteasome subunit abundance and chronic UPR activation, as expected. Furthermore, we observed that untreated *∆rpn4* cells accumulate proteins normally degraded by ERAD, upregulate proteasome assembly factors, mount a constitutive heat shock response and attenuate ribosome biogenesis, illustrating a broad adaptive response (see the second paragraph of the aforementioned subsection and new Figure 4D). Finally, we found that, surprisingly, *∆rpn4* cells still elevate proteasome subunit abundance during ER stress (subsection “Rpn4 and the UPR cooperate to counteract ER stress”, last paragraph and new Figure 5—figure supplement 2). However, this response is insufficient to reach the levels seen in wild-type cells.

Reviewer #1:

[…] There two points of critique that I feel should be addressed:

The authors present a strong case that in the unbuffered state (e.g. tunicamycin-treated ∆hac1 yeast) activation of Rnp4 arises by the presence in the cytosol of proteins that would otherwise find their way into the ER. This begs the question if the buffering of the growth defect arising from Rnp4 over-expression reflects predominantly the role of the proteasome in alleviating this cytosolic problem, or if the presumptive enhanced proteasomal function of Rpn4(+) cells also reflects a cross compartment effect, whereby the Rpn4(+); E2-treated ngCPY*; ∆hac1 cells also experience lower levels of ER stress; lower than the comparison group in which Rpn4 is not over-expressed? It may be possible to gage the level of unfolded protein stress in the ER by monitoring the state of Ire1p activity. I'm not sure if it is possible to monitor endogenous Ire1p phosphorylation in budding yeast, but perhaps this question can be answered by introducing an inert Hac1 splicing reporter into the experimental system.

Using reporters for the activity of Ire1, Hsf1 and the stress response transcription factors Msn2/4, we now show that *RPN4* overexpression can improve protein folding homeostasis in both the ER and the cytosol of ER-stressed cells (subsection “Rpn4 is upregulated by and protects against mislocalized secretory proteins”, last paragraph and new Figure 7E). This finding supports the proposed cross-compartment response to ER stress.

The authors speculate on the possibility that buffering observed by Pdr1 and Ssz1 over-expression may be channelled through their role in raising the levels of Rpn4, thus providing a unifying mechanism of action for the top hits in this screen. Can this notion be tested? Is the buffering by Pdr1(+) and Ssz1(+) of E2-treated ngCPY*; ∆hac1 cells eliminated by ∆Rpn4?

We clarified this point by generating strains in which the *RPN4* gene was placed under the control of the constitutive *CYC1* promoter and thus uncoupled from its normal regulation. Overexpression of *PDR1, SSZ1* or *YAP1* still promoted ER stress resistance in this background. Therefore, Pdr1, Ssz1 and Yap1 can act through mechanisms unrelated to *RPN4* gene activation (subsection “A screen for genes promoting ER stress resistance in UPR mutants”, second paragraph and new Figure 3A). We also tried to do similar experiments in *∆hac1 ∆rpn4* cells but their extremely poor growth even in the absence of exogenous ER stressors (see Figure 3B and 3C) rendered these attempts uninformative.

Reviewer #2:

[…] Overall, this is an interesting manuscript addressing important questions related to the coordination of stress-pathways in response to diverse insults. The experiments are generally well performed and the conclusions are supported by the experiments. However, I have a few specific comments primarily pertaining the relationship between RPN4 activation and specific biologic functions during ER stress that should be addressed prior to publication. I outline these comments below.1) The authors suggest that other genes identified in their screen (e.g., PDR1, SSZ1, and YAP1) promote cellular growth of hac1-deficient cells in the presence of ER stress through activation of RPN4. If this is the case, one would predict that overexpression of these genes would not rescue growth in rpn4-deficient yeast incubated with increasing amounts of ER stress. This would provide further evidence that these genes function in the screen through the upregulation of RPN4, as suggested. Alternatively, this could function through alternative mechanisms. Either way, it would be interesting to define the specific dependence of gene rescue from these different hits on RPN-4.

We clarified this point by generating strains in which the *RPN4* gene was placed under the control of the constitutive *CYC1* promoter and thus uncoupled from its normal regulation. Overexpression of *PDR1, SSZ1* or *YAP1* still promoted ER stress resistance in this background. Therefore, Pdr1, Ssz1 and Yap1 can act through mechanisms unrelated to *RPN4* gene activation (subsection “A screen for genes promoting ER stress resistance in UPR mutants”, second paragraph and new Figure 3A). We also tried to do similar experiments in *∆hac1 ∆rpn4* cells but their extremely poor growth even in the absence of exogenous ER stressors (see Figure 3B and 3C) rendered these attempts uninformative.

2) The authors suggest that the increased UPR afforded by ER stress in rpn-4 deficient yeast results from the ER buildup of misfolded ERAD substrates. This is probably true, but it would be good to see the increased ER accumulation of these proteins to confirm this model.

As part of our proteomic analyses (see point 4 below), we found that deletion of *RPN4* increases the abundance of several proteins that are involved in sterol synthesis and constitutively turned over by different ERAD pathways (subsection “Rpn4 and the UPR cooperate to counteract ER stress”, second paragraph and new Figure 4D). This observation supports the model that lack of Rpn4 activity leads to a buildup of ERAD substrates and subsequent UPR activation. Importantly, the alternative explanation that lack of Rpn4 activity causes UPR activation by a different mechanism and in this way upregulates expression of the above ERAD substrates is unlikely because the genes encoding these ERAD substrates are not regulated by the UPR.

3) The authors suggest that accumulation of ER proteins through either increased ERAD or impaired translocation is responsible for the increased RPN4 activity observed during ER stress. They do a nice job of testing the impaired translocation using the sec65 mutant. However, it would be interesting to define the importance of ERAD in this process using hrd-1 deficient yeast. If ERAD is required, one would expect that overexpression of RPN4 should not rescue growth under these conditions. It would be nice to complete the study by testing this alternative mechanism of increased cytosolic protein load afforded by increased ERAD.

It appears that we misled the reviewer by our inaccurate portrayal of the role of *HRD1* in ERAD. While important, Hrd1 is not ‘essential’ for ERAD as we had phrased it, because several ERAD pathways exist even in yeast (mediated by the ubiquitin ligases Hrd1, Doa10 and Asi1/2/3). Therefore, the suggested epistasis experiment of completely blocking ERAD and asking whether now *RPN4* overexpression no longer rescues growth is difficult to do and outside the scope of this revision. However, the fundamental point raised by the reviewer remains valid and it will be interesting to determine whether the increased cytosolic protein folding load that activates and is mitigated by Rpn4 arises from impaired translocation, increased ERAD, or both.

We originally included *HAC1* splicing in *∆hrd1* cells merely as a reference point for the *HAC1* splicing seen in *∆rpn4* cells. Since the qPCR data (Figure 4B and C) and proteomics data (new Figure 4D) clearly show constitutive UPR activation in *∆rpn4* cells, we would like to avoid any confusion and have removed *∆hrd1* cells from Figure 4A. The figure’s conclusion, i.e. that there is elevated *HAC1* splicing in *rpn4∆* cells, is unaffected by this change.

4) The idea is that increases in RPN4 coordinate with the UPR by upregulating proteasome biogenesis (i.e., increasing proteasome capacity). However, increased expression of proteasome subunits and measures of proteasome activity are not provided (although they do show modest increases in the degradation of cytosolic proteins in cells following RPN4 overexpression). If RPN4 is protective during ER stress through increased proteasome biogenesis, shouldn't there be a measurable increase in proteasome capacity? Also, shouldn't there be an increase in proteasome subunit expression? This type of information should be provided to support the overall model proposed in panel 7F.

To address this point, we performed proteomic analyses of untreated and tunicamycin-treated wild-type and *∆rpn4* cells. As anticipated from the activation of Rpn4 by ER stress, the expression levels of proteasome subunits and proteasome assembly chaperones rose in tunicamycin-treated wild-type cells (subsection “Rpn4 and the UPR cooperate to counteract ER stress”, last paragraph and new Figure 5F). A comparison of wild-type and *∆rpn4* cells at steady state showed a reduction in proteasome subunit abundance and chronic UPR activation, as expected. Furthermore, we observed that untreated *∆rpn4* cells accumulate proteins normally degraded by ERAD, upregulate proteasome assembly factors, mount a constitutive heat shock response and attenuate ribosome biogenesis, illustrating a broad adaptive response (subsection “Rpn4 and the UPR cooperate to counteract ER stress”, second paragraph and new Figure 4D). Finally, we found that, surprisingly, *∆rpn4* cells still elevate proteasome subunit abundance during ER stress (subsection “Rpn4 and the UPR cooperate to counteract ER stress”, last paragraph, and new Figure 5—figure supplement 2). However, this response is insufficient to reach the levels seen in wild-type cells.

5) Along the same lines as #3, does ER stress induce proteasome biogenesis in mammalian cells? There are lots of transcriptional profiles available, and it would be interesting to see if this model is conserved in higher eukaryotes. If not, are there other mechanisms involved that perform a similar function such as the iRHom regulation during ER stress in flies reported by Lee et al., 2015?

Prompted by this query and also the comment of reviewer #1 above, we expanded the section on mammalian cells in the Discussion and included the Lee et al. paper mentioned above (Discussion, fifth paragraph). We use Nrf1 and the iRhom1-regulated PAC1/2 as examples to illustrate how ER stress can induce proteasome biogenesis in mammals. A systematic analysis of transcriptional profiles would certainly be interesting but is outside the scope of this revision.

6) The authors show that they can suppress the transcriptional upregulation of RPN4 by deletion of MSN2/4 and removal of the YRE and HSE from the rpn-4 promoter. Can they use this approach to define the relative importance of posttranscriptional and transcriptional regulation of RPN4 during ER stress? It would be interesting to define the specific dependence of yeast protection against ER stress on these two aspects of RPN4 regulation.

It would indeed be of interest to understand the relative contribution of the post-transcriptional and transcriptional regulation of Rpn4 to ER stress resistance. However, since it remains to be established how Msn2/4 activate the *RPN4* promoter (also see response to the first comment of reviewer #3 below), the only way currently available to block the transcriptional induction of *RPN4* involves deletion of Msn2/4, which will by itself cause stress sensitivity. Hence, this question cannot be addressed cleanly at this point.

Reviewer #3:

[…] The only aspect of the paper that is wanting is the end, where the authors bash the RPN4 promoter and use analog sensitive PKA and msn2/4∆ strains to attempt to define the signals that activate RPN4. The approach here is a bit muddled and as such the conclusion is not clear. To bring this home, I would recommend doing a clean necessary/sufficient scan of the TF binding sites in the RPN4 promoter. Start with 1kb upstream of RPN4: there are 6 STREs, 2 PDREs, 1 YRE and 1 HSE (I am happy to send my analysis of where the STREs are, as apparently the authors could not find them). In addition to the PDRE, YRE and HSE mutants the authors have, order a GBlock lacking the STREs and do the RT-qPCR. This will complete the necessity set. For sufficiency, just order 4 Gblocks that have only the 6 STREs, 2PDREs, the YRE or the HSE. Then add Tm and induced ngCPY* and measure RPN4 expression and growth in this set. The PKA/msn2/4 mutants, while clever, don't make the point as clearly.

Direct communication with the reviewer established that there is no canonical STRE in the *RPN4* promoter, so the suggested experiments cannot be done.